# Interwoven traditions in Bell Beaker metallurgy: Approaching the social value of copper at Bauma del Serrat del Pont (Northeast Iberia)

Julia Montes-Landa[1]*, Mercedes Murillo-Barroso[2], Ignacio Montero-Ruiz[3], Salvador Rovira-Llorens[4], Marcos Martinón-Torres[1]

1 Department of Archaeology, University of Cambridge, Cambridge, United Kingdom, 2 Departamento de Prehistoria y Arqueología, Facultad de Filosofía y Letras, Universidad de Granada, Granada, Spain, 3 Instituto de Historia-CSIC, Madrid, Spain, 4 Museo Arqueológico Nacional, Madrid, Spain

* jm2219@cam.ac.uk

**Data Availability Statement:** All relevant data are within the manuscript and its Supporting Information files.

## Abstract

Debates on early metallurgy in Western Europe have frequently focused on the social value of copper (between utilitarian and symbolic) and its purported role in the emergence and consolidation of hierarchies. Recent research shows that generalisations are increasingly untenable and highlights the need for comparative regional studies. Given its location in an intermediate area, the early metallurgy of Northeast Iberia provides an interesting case in point to explore the interaction between the well-characterised traditions of southern Iberia and southern France during the 3rd and 2nd millennia BCE. Here the analytical study of seven Bell Beaker (decorated and undecorated) vessels reused as crucibles at Bauma del Serrat del Pont (Tortellà, Girona) are presented. We employed pXRF, metallography, SEM-EDS and lead isotope analyses. The results show evidence for copper smelting employing a remarkable variety of ore sources, including Solana del Bepo, Turquesa and Les Ferreres mines, and an extra unknown area. The smelting vessels were manufactured using the same clay, which contained both mineral and organic inclusions. Our results are discussed with reference to all the evidence available for metals and metallurgy in the Northeast, and more broadly in comparison to southern Iberia and southern France, with special emphasis on issues of production organisation and social complexity. Taken together, our results support the notion that copper metallurgy played a predominantly utilitarian role in Bell Beaker societies and highlight idiosyncratic aspects of the metallurgical trajectory in the Northeast. Differences between territories challenge unilinear explanations of technological and social development after the introduction of metallurgy. Separate trajectories can only be explained in relation to area-specific socio-cultural and environmental factors.

**Funding:** This paper has been supported by the project "Metal y Ámbar II: Circulación de bronce y ámbar en el Sureste peninsular durante la Edad del Bronce" (PID2019.108289GB.I00 / SRA – State Research Agency / 10.13039/501100011033) funded by the Spanish Government and with MMB as PI. The part of the analytical work of this paper that was carried out by JML (SEM-EDS and metallography) was supported by the Arts and Humanities Research Council-DTP program and the Cambridge Trust (Vice-chancellor's Award), that jointly fund her doctoral research. Funders had no role play in study design, data collection and analysis, decision to publish or preparation of the manuscript.

**Competing interests:** The authors have declared that no competing interests exist.

## Introduction

Bauma del Serrat del Pont (Tortellà, Girona; La Bauma henceforth) is a key site for understanding the origins of copper extractive metallurgy in the Iberian Northeast, as it presents the earliest assemblage related to metallurgical activities in the area. Its location between two distinct regional technological traditions (Southern Iberia and Southern France) stimulates questions about the parallel influence of these traditions in this intermediate area during the end of the 3$^{rd}$ millennium BCE, which can be approached through detailed microstructural and compositional analyses of the crucibles recovered.

The differential socio-cultural dynamics operating in southern Iberia, northeastern Iberia and southern France during the late 3$^{rd}$–early 2$^{nd}$ millennium BCE resulted in differential paths towards social complexity. These paths are to some extent reflected in the different role of copper within these societies, as shown by traditional approaches to finished metallic objects. By characterising copper production debris at La Bauma and discussing these finds in relation to other evidence from the Northeast and beyond, we aim to contribute to the narrative of non-linearity towards social complexity. Our study shows that the social value of copper is also interlinked with its production and the organisation of these activities, which ultimately contributes to a better understanding of Chalcolithic societies in the Northeast.

## Technological and social characterisation of early metallurgy in the Northeast: Contextualising Bauma del Serrat del Pont

### Chalcolithic metallurgy in the Northeast: between two technological traditions

La Bauma yields the earliest extractive metallurgy evidence in northeastern Iberia. However, in southern Iberia and southern France, the two closest regions with Chalcolithic copper technology, copper metallurgy was practiced since earlier times according to separate technological traditions.

The earliest evidence of copper extractive metallurgy in the whole Iberian Peninsula, dated to the 5$^{th}$ millennium BCE, comes from Cerro Virtud (Almería), which supports an independent discovery of metallurgy in this area [1–3]. It is only from the 4$^{th}$ and especially the 3$^{rd}$ millennium BCE when consistent evidence supports sustained production in the Southeast (e.g. Las Pilas, Santa Bárbara, Almizaraque, Ciavieja) and Southwest (e.g. Cabezo Juré, Valencina de la Concepción), including Portugal (e.g. Zambujal, Vila Nova de São Pedro) [4–12].

Analyses of assemblages from these sites show that Iberian Chalcolithic communities conducted a simple crucible-based metallurgy developed within *vasijas-horno* ('crucible-furnaces'): shallow open common pottery vessels with rounded bottoms; very different from other Neolithic/Chalcolithic crucibles from Europe and the Near East, and from modern cup crucibles [13]. Local copper carbonates and oxides were smelted under variable reducing conditions that usually produced an incomplete reaction of the charge [14–17]. These processes generated no or little slag and when they did, this was crushed to recover the metallic prills trapped. As is common in early metallurgy, a metallic mass of the desired shape was cast after re-melting the prills, and was later hammered [17, 18]. Contrary to other parts of Europe, annealing was not extensively used in Iberia until later times, so objects were usually cold hammered [10].

Early French metallurgy is believed to have spread from northern Italy (Rinaldone) [19]. Although initially only items related to high social status penetrated the Paris Basin [20], at the beginning of the 3$^{rd}$ millennium BCE, extractive metallurgy emerged at the Cabrières-Peret district (Languedoc) (e.g. Roque-Fenestre, La Capelle du Broum, Valat-Grand and Pioch-

Farrus A-448, and Neuf-Bouches and Petir-Bois mines) [21–25]. La Capitelle du Broum is the earliest specialised Chalcolithic metallurgical site in France [22].

Contrary to the case of southern Iberia, smelting processes here took place directly in open pits with no smelting crucibles. The slag was similarly crushed to recover metallic prills, which were remelted into an amorphous mass in a non-movable, elongated 'ingot crucible' or *lingotera*. This metallic mass was then hammered into the desired shape [26]. Annealing was practiced by these communities [22]. They used local unroasted polymetallic fahlore ores with remarkable amounts of antimony and silver, and minor presence of arsenic, zinc, lead and low iron [14, 22, 25]. Most of the metallic objects found reflect this composition, although some of them are purer [22]. The latter has led to suggestions that co-smelting of oxidic and sulphidic ores could have been conducted [23, 26]. Interestingly, smelting crucibles only occur in France from the middle of the 3rd millennium BCE, an innovation related to the Bell Beaker phenomenon (e.g. Al Claus, Travers des Fourches, Serrer de Boidons, Courondes and Bousquets, La Republique II, L'anse de la République and Saut Châteauneuf-les-Martigues) [14, 23, 24, 27, 28].

The Northeast can be seen as an area in-between two technological traditions that differ in three key aspects: use of smelting crucibles *vs*. open pits, smelting oxidic *vs*. sulphidic ores, and use or not of annealing. The geographic barrier of the Pyrenees might lead to the assumption that the operating dynamics in the Northeast did not extend to the other side of these mountains. However, during the late 4th and 3rd millennia BCE, strong cultural connections clearly linked this area to southern France (see The social value of Chalcolithic copper in the Western Mediterranean); the Pyrenees were not necessarily perceived as a barrier. Thus, discussion on the origins of the metallurgical know-how in the area, and particularly whether it derives from France or Iberia, is still open. The overall evidence discussed by Soriano Llopis [29] supports that extractive metallurgy know-how spread from southern France, based on closer chronologies between southern France and northeastern Iberian sites, and absence of 3rd millennium BCE extractive evidence (crucibles, slags) at the regions between southern Iberia and the Northeast. However, early extractive metallurgy evidence in the Northeast (see Evidence of crucible metallurgy in the Northeast: Bauma del Serrat del Pont and its contemporaneous assemblages) is consistent with crucible metallurgy, thus reminiscing Iberian technological dynamics too. The introduction of crucible metallurgy in France in the middle of the 3rd millennium BCE has been interpreted as an innovation transmitted from Iberia [23, 29]. In this way, the crucible assemblage from La Bauma, the earliest site with evidence of copper processing in the Northeast, can be considered to some extent a bridge between these two technological traditions: know-how introduced from France, adapted to crucible metallurgy technology of southern Iberian origins.

La Bauma has the earliest radiocarbon dated contexts with extractive metallurgy evidence in the area (2878–2479 cal. BCE, Table 1). The stratigraphic sequence comprises strata from Mesolithic to Roman times [30]. Levels II.3 (Early Bronze Age, EBA), II.4, II.5 and III.1 (Chalcolithic) contained the majority of the archaeometallurgical evidence. Level II.5 was characterised as a metallurgical production context. The rest of Chalcolithic and EBA occupations were habitational contexts (see Materials analysed and their archaeological contexts). The 65 crucible sherds found belong to a minimum of 14 undecorated vessels and five decorated Bell Beaker pots. On Alcalde *et al*. [31] drawings of some of these vessels are available. The use of decorated pots for metallurgical operations is not a unique feature of La Bauma. Other Bell Beaker decorated crucibles have been found at El Ventorro (Madrid province), Sont Matge (Mallorca), El Acebuchal (Seville province) and Travers de Fourches (Hérault) [32–35]. At La Bauma, two tuyères and a melting residue were also found in level II.3 (EBA) and one more tuyère and another melting residue in level II.5 (Chalcolithic), together with a potential

**Table 1. Samples analysed, their stratigraphic context, $^{14}$C calibrated dates, analyses conducted and summary of results.** Samples are arranged following stratigraphic order, from the most superficial level at the top and the deepest one at the bottom.

| Sample reference | | Decorated | Context | $^{14}$C dates | CA code | SEM | Ore | LIA | Proposed provenance |
| Square | Ex. ID | | | | | | | | |
|---|---|---|---|---|---|---|---|---|---|
| H12 | 29inf | | II.4 | 2891–2466 cal. BCE | CA190131 | x | Ca | x | Solana del Bepo |
| E13 | 30inf | | II.5 | 2917–2576 cal. BCE | CA190127 | x | P | x | Group 2 |
| G11 | 32sup 143 | x | | | CA190130 | x | Ca | | |
| E11 | 32sup 136 | | II.5/III.1 | No data | NA | | | x | Group 2 |
| F12 | 32sup 144 | x | III.1 | 2877–2291 cal. BCE | CA190126 | x | Ca | | |
| F11 (= G12) | 32sup (30sup) | | III.1 (II.4) | 2877–2291 cal. BCE (2891–2466 cal. BCE) | CA190128 | x | P | x | Les Ferreres |
| G10 | 68 | | III.1/III.2 | 2878–2479 cal. BCE | CA190129 | x | Ca | x | Turquesa (/Solana del Bepo) |

Ex. ID = excavation ID; CA code = Cambridge Archaeology laboratory code; NA = not applicable; Ca = Ca-rich ore; P = Pure ore.

grinding stone. An arrowhead was found in III.1, an awl in II.4 (both Chalcolithic), and a fragment of another awl, and an undetermined elongated item in II.3 [31].

La Bauma is also the earliest tin bronze production site in Iberia, as indicated by previous ED-XRF analyses on a Chalcolithic vessel (4.63wt%Sn), melting residue (7.58wt%Sn) and arrowhead (7.09wt%Sn). The two metallic finds are very close to the perceptual threshold for copper and tin alloying (~6wt%Sn) [36, 37], which raises questions about the intentionality of this alloy potentially produced by co-smelting naturally mixed Cu-Sn ores. Interestingly, such mixed ores occur at the nearby mines (*ca.* 18km away) of Les Ferreres (Rocabruna) and Can Manera (Albanyà) [31]. Although we lack isotopic data, ED-XRF data on ores found at the site suggests that they might come from two different sources–one purer, and the other one containing tin. Later EBA evidence (2566–2034 cal. BCE) from the site confirms full development of bronze metallurgy (*i.e.* alloying of copper ores with cassiterite) with tin levels over 23wt%Sn in a melting residue and an awl [31].

The analyses of Chalcolithic samples from La Bauma highlight that the beginning of extractive metallurgy in the Northeast involved both copper smelting and bronze production through a potential natural alloying process. This contrasts with southern Iberia, where pure copper and/or arsenical copper was produced. In France, some analyses of early daggers also point towards an early introduction of bronze metallurgy [38].

Alcalde *et al.* [31] presented very promising results based on ED-XRF analyses of the materials from La Bauma, which were complemented by some metallographic analyses by S. Rovira (unpublished). Despite the absence of more detailed analyses, it has been accepted that the crucibles from La Bauma were used for smelting. However, it is possible that these vessels could have been used for re-melting scrap metal. The differentiation of smelting from melting crucibles based on macroscopic features such as the thickness of the attached slag layers is not always conclusive [17, 39], especially in early metallurgy contexts: microstructural analyses are necessary.

Moreover, a more thorough characterisation of the crucible charge becomes an important question for understanding innovation dynamics and organisation of production. Beyond discerning the use of oxidic/sulphidic ores, provenancing the copper smelted/melted would add to our knowledge of social dynamics behind the exploitation and distribution of raw materials.

Finally, considering that decorated Bell Beaker vessels were used for metallurgical operations, it is important to characterise these ceramics so they can be compared to vessels from other Iberian areas, including later sites in the Northeast and southern France, to improve our understanding of the evolution of the crucible-making technology in the area.

## The social value of Chalcolithic copper in the Western Mediterranean

It is necessary to contextualise this technological study with reference to the social role and value of copper in the Western Mediterranean: southeastern Iberia, northeastern Iberia and southern France.

Based primarily on evidence from the South, it has been argued that copper was not a particularly valuable raw material in Iberia during Chalcolithic times (*ca*. 3200–2200 cal. BCE). Unlike amber, ivory or ostrich eggshell–usually found in funerary contexts–the value of copper seems to be more limited, as it is equally found in funerary and habitational contexts. Employed primarily for tools and tool-weapons (*i.e.* items that could perform both functions such as an axe), the metal was never used as an ornament and it is absent in the wealthiest burials. The low social relevance of copper can be a consequence of the wide availability and accessibility of its ores, the simplicity of the extraction process, and the domestic low-scale character of this activity [3, 13, 16].

When comparing this pattern to the 3[rd] millennium BCE in the Northeast, some differences emerge. Between *ca*. 3300–2250 BCE, the so-called Late Neolithic-Chalcolithic groups–commonly referred to as Veraza–populated this area. This cultural complex developed on both sides of the Pyrenees and it is mainly recognised by its shared material culture. Here, highly mobile communities practiced agriculture and stockbreeding but lacked smelting know-how. Half of the metallic items recovered from these groups (beads, ornamental small sheets, flat axes and tongue daggers) are ornaments (including copper and gold items), and the other half are tool-weapons, even though ornaments are commoner in other raw materials (schist, lignite, steatite, bone, stone) [29]. Metallic items were deposited in traditional communal burials rather than prominently associated to individuals. Thus, overall, the evidence suggests that copper held a modest social or symbolic value in the Northeast, even though a more significant one than in the South [29]. The different symbolic treatment of copper in the Northeast at the beginning of the 3[rd] millennium BCE may be related to the absence of smelting know-how, which rendered copper more exotic and appreciated as an ornamental raw material.

The presence of copper ornaments in the Northeast must be associated to southern France dynamics, not only because their composition can be related to their characteristic ores [29] but also because these types of items occur at the Paris Basin since the end of the 4[th] millennium BCE, and were manufactured at Languedoc since early 3[rd] millennium BCE. During this millennium, the social value of copper in France shows parallels with the Northeast, with copper ornaments taking an important symbolic role and equated to other raw materials. When tool and weapon types occur in early contexts, they lack traces of use and are clearly related to symbolic rather than utilitarian functions [20]. Interestingly, however, only at the Cabrières district (Hérault) metal production was in fact associated to social changes towards complexity, in connection with habitat patterns and organisation of production to mine and smelt the local copper ores. In other areas such as Tarn-et-Garonne, metallurgy was integrated within domestic sporadic activities and the production output was more reduced [20].

The last part of the 3[rd] millennium BCE coincides with the start of Bronze Age (2250–1550 cal. BCE) in the Southeast, a period that will bring important social changes. During the BA, fortified settlements proliferated parallel to the development of elites and single burials. Amber, ostrich eggshell and other exotic raw materials were no longer used, and metallic ornaments (including copper, silver, gold, and bronze, especially after 1800 cal. BCE) together with weapons, proliferate in tombs as a status symbol of the new emerging elite. Annealing was also introduced in the *chaîne opératoire* at this moment, which facilitated the manufacture of copper ornaments for elites. The necessity of weapons for social coercion would have also contributed to raise the social value of copper. This process of social stratification correlated with the

change in the social meaning of copper (even though causality has not been conclusively established), giving rise to the Argaric society in the Southeast [13, 16, 40].

Things were different in the Northeast during the late 3<sup>rd</sup>–early 2<sup>nd</sup> millennium BCE. Around 2700 BCE, the Bell Beaker phenomenon emerged in the area, with new traditions co-existing with the Late Neolithic-Chalcolithic ones already described. Bell Beaker peoples inhumated a smaller number of individuals than their contemporaries, and bodies are now clearly individualised within communal tombs. There are also cases of single and double burials, including individuals of both sexes and different age ranges, perhaps pointing to inherited social recognition [29].

All copper objects manufactured by Bell Beaker communities were tools or weapons. No copper ornaments have been found. Metallic palmella points and other types of arrowheads were incorporated to the archaeological repertoire. However, the same types of object are still more common in other raw materials such as stone. Most of the metallic objects recovered appeared in funerary contexts and are related to specific individuals. This might indicate a higher social value for them than for other items typical of the so called "Bell Beaker pack". However, unlike gold ornaments, it is not likely that copper production was focused on funerary consumption, as all buried objects were heavily used. They probably had both a utilitarian and social (possibly identitarian) function. The extent of the latter is difficult to establish with the current evidence, especially when considering the sporadic low-scale production of this metal in the Northeast (see Evidence of crucible metallurgy in the Northeast: Bauma del Serrat del Pont and its contemporaneous assemblages). In any case, copper appears to have been more important for Bell Beaker communities than for other contemporaneous Chalcolithic groups, and perhaps it was related to the beginning of social stratification–possibly based on cattle control. In this way, the advantageous properties of copper for executing coercion might have been appreciated by the emerging influential social sector [29].

Metallurgy was one of many other activities (e.g. flint knapping, bone working, vegetal processing, etc.) carried out by the small Bell Beaker communities of the Northeast. These groups, although broadly fixed in the territory, were considerably mobile. They practiced itinerant stockbreeding and agriculture, and lived in caves and natural shelters (see Evidence of crucible metallurgy in the Northeast: Bauma del Serrat del Pont and its contemporaneous assemblages) that were also used for funerary purposes [29]. This contrasts with the open habitats of southern France. However, there were local exceptions to this pattern such as the Vallés area, where hut remains of more sedentary communities have been discovered, for example at Vapor Gorina site [41].

Thus, there is a change in the social value of metal in the Bell Beaker groups of the Northeast compared to their previous and contemporary Veraza neighbours. This shift can be linked to very incipient social changes that are broadly comparable to those seen in southeastern Iberia during the beginning of EBA. However, the crystallisation of social hierarchies in the Northeast had to wait until Iron Age (IA). The social conditions necessary for the development of an elite interested in quickly modifying the social meaning of copper were still not present in the Northeast at this time.

Interestingly, these two patterns towards social complexity differ from the panorama in southern France at the end of the 3<sup>rd</sup> millennium BCE. After the metallurgical development seen at the Cabrières district, there was a decrease in metal production and consumption in Languedoc. This activity was then moved until the middle of the 2<sup>nd</sup> millennium BCE to the Hautes-Alpes district, where copper extractive metallurgy was technologically improved. Mille and Carozza [20] have attributed this change to EBA social dynamics in northern Italy.

Thus, the role of copper followed very different trajectories in relation to supporting social complexity. While in southeastern Iberia there was a progressive transformation of its value

favoured by the emerging elites, in Languedoc, the initial higher status of copper–which had potential for supporting emerging elites as already structured habitat patterns–was dismantled completely at the end of the 3$^{rd}$ millennium BCE. In the Iberian Northeast, conversely, the initial low value but symbolic use of copper progressively incorporated functional use. However, here there were no well-defined emerging elites to which a change in the social value of metal might have specially benefited. Overall, a comprehensive analysis of the contextual social, economic and environmental dynamics is necessary if we want to understand the role of copper in the path towards social complexity–a path with multiple diversions and with no linearity.

Discussion of the social meaning of metal has relied mostly on the types of manufactured items and their depositional contexts. Technological analysis of crucibles and (s)melting assemblages such as those from La Bauma can contribute to this topic from a different perspective. Discerning between smelting and melting is critical to understand the technological knowledge during the earliest metal processing in the Northeast. While absence of smelting know-how might have induced a higher value of copper as a scarce foreign resource, smelting knowledge might have facilitated access to metallic items, thus reducing their social value. Evidence for recycling operations could also reinforce to some extent a higher value for this metal.

Corroborating a local provenance of the copper ores can also contribute to support considerations on the social value of copper. For instance, proof of this can be found in the differences observed in habitat patterns at the Cabrières and Tarn-et-Garonne districts during Chalcolithic times. In the former area, sites were located considerably close to the mines, as they were mainly dedicated to metallurgical activities. In the latter area, although sites were still close to local mines, they were more distanced, which corresponds to a more domestic character of copper metallurgy [20]. Finally, characterising the crucible pastes could help assess if the production of these decorated pots also reflected a specific recipe geared towards metallurgy.

## Evidence of crucible metallurgy in the Northeast: Bauma del Serrat del Pont and its contemporaneous assemblages

Apart from La Bauma, there are other contemporary sites with extractive metallurgy materials in the Northeast (Fig 1). Copper mining evidence is concentrated at El Priorat mines, at Montsant and Molar-Bellmunt-Falset mining districts [42, 43]. The characterisation of many of the contexts with extractive evidence as well as the absolute dating of many of these sites is problematic. Some of them have been radiocarbon dated to the Chalcolithic, but usually only one date is available for the whole stratum/site. That is why, for many sites, a broader ascription to the second half of the 3$^{rd}$ millennium BCE (the so referred to as Chalcolithic-EBA transition) is commonly used in the literature, based on typological studies of related materials. This section compiles all evidence that can be ascribed to the 3$^{rd}$ millennium BCE.

Moving from North to South, Cova Joan d'Ós (Tartareu) is considered a Chalcolithic site with unclear stratigraphy [44], which has prevented researchers from narrowing down its functional use. However, habitat and a burial phases were suspected in different moments [45, 46]. A bronze dagger–not related to the burial–was found at the bottom of the cave, together with a flat bronze (9.45wt%Sn) axe from a superficial context, and an amorphous bronze (8.17wt%Sn) mass [44, 45, 47]. The dagger has been ascribed to early LBA [45]. Tin content in the semi-processed mass (8.17wt%Sn) is comparable to those in samples ascribed previously to bronze alloying through the smelting of polymetallic ores at La Bauma (see Chalcolithic metallurgy in the Northeast: between two technological traditions), but further evidence is needed to confidently date this find.

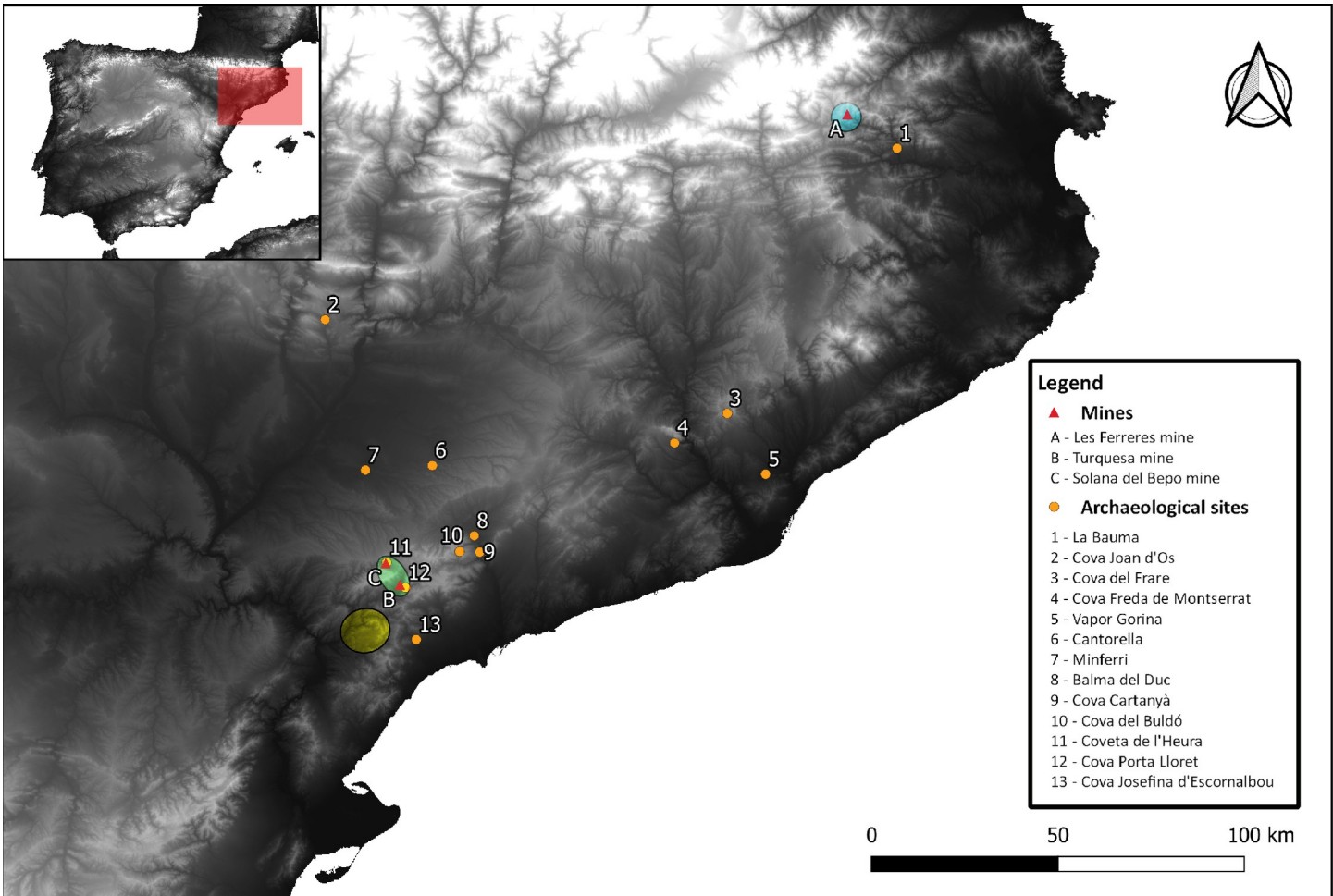

**Fig 1. Map of the Northeast with key sites mentioned in this paper.** (Green circle) Montsant mining district. (Yellow circle) MBF mining district. (Blue circle) Les Ferreres mining district. (Raster model: STRM DEM from NASA).

Cova del Frare (Matadepera) presents materials dated between the Early Neolithic and the EBA. LBA and IA-I materials were also found. All contexts were characterised as sporadic occupations except layer 4, which was a Neolithic (Veraza) burial [48–50]. Bell Beaker sherds with copper residues–as confirmed by pXRF–were found in layer 3 (2040 cal. BCE) together with a possible copper axe. Other metallurgical materials from later periods comprise a ternary bronze ring, a plano-convex copper ingot and a fragment of a second one [49].

Cova Freda de Montserrat (Collbató) is an Early Neolithic Cardial site (6th millennium BCE) used as a cattle refuge. Burials were also conducted in it at unspecified later times, and materials from the Iberian Period were also recovered [51]. The bronze (7.67wt%Sn) plano-convex ingot found at the bottom of the cave [49] is unlikely related to Neolithic times, as no evidence of metallurgy exists in the Northeast for such early dates, especially containing tin. Given the chronological range of the materials found and the absence of information on the archaeological context where it was found, it is problematic to broadly date it [52]. The absence of ingot circulation in Iberia until the LBA also makes unlikely a Chalcolithic or EBA ascription to this ingot [53].

Vapor Gorina (Sabadell) is a Chalcolithic Bell Beaker site. Here, six nodular slags and an awl fragment were found within the remains of a perishable hut [41]. Compositional analyses and lead isotopes analyses (LIA) of two copper prills and the awl indicated use of a copper ore with minor arsenic and lead that could not be related to El Priorat mineralisations [54].

Moving on to Tarragona province, this area has been studied more in-depth, and it yields the richest in archaeometallurgical evidence. Materials from Balma del Duc (Montblanc), dated between Chalcolithic and Middle Bronze Age (MBA), were related to a temporary habitat. No burial has been found [55]. Rovira and Ambert [23] classified the crucible from this cave as Chalcolithic, although it comes from a poorly defined chrono-stratigraphic context [56], which led other authors to date it to the EBA [29]. LIA related the copper ores processed to Turquesa mine (Cornudella de Montsant) [54, 57, 58]. Compositional analyses indicated use of a copper ore with occasional tin impurities (0.16wt%Sn) [29, 54].

Cova del Buldó (Montblanc) has been used as a communal funerary context in three different areas. A crucible dated between Chalcolithic-EBA was found related to two infants [29, 52]. Minor amounts of tin (0.95wt%Sn) characterised the copper ore used. LIA indicated that the ore might have come from Turquesa mine, notwithstanding some disagreements in specific ratios [29, 54, 56].

A crucible related to copper metallurgy by pXRF was found at Cova Cartanyà (Vilavert), a Chalcolithic-EBA site with Bell Beaker materials [23, 52, 59]. The cave was sporadically inhabited and no funerary remains were found. However, some human bones were dispersed at different spots of the main chamber of the cave [60]. LIA could not relate the crucible residue to El Priorat [54, 61].

Cova de l'Heura (Ulldemolins) was firstly occupied at the end of Neolithic-Chalcolithic as a shelter and a flint arrowheads manufacturing area. During the early 3$^{rd}$ millennium BCE it served as a collective burial place and during the EBA it was used for metallurgical work [23, 56, 62, 63]. One crucible was recovered, together with fragments of copper ore and melting droplets, a bronze bead, a lead bead, an arsenical copper awl, an elongated bronze item, and a metallic ingot-like mass. While the lead bead and the copper awl are ascribed to late Neolithic-Chalcolithic, the bronze bead and the crucible are probably from the EBA [56]. LIA indicated that the copper of the awl comes from Turquesa mine while the crucible charge can be linked either to Barranc Fondo (Cornudella de Montsant) or Solana del Bepo (Ulldemolins) mines. However, minor elements do not perfectly fit this interpretation [54, 56–58]. Provenancing the mineral relicts was problematic as no perfect fit in relation to minor elements was found with El Priorat mines [54, 57, 64].

Materials from Cova de Porta-Lloret (Siurana) can be dated from Chalcolithic to BA with occasional items from LBA and IA. Bell Beaker materials were found at the bottom of the cave together with later material culture. The cave lacks clear stratigraphy and no evidence of human remains was found. A melting residue (2.85wt%Sn/1.74wt%Sn in separate analyses) together with a copper ingot were recovered. Given the long chronology of the site, it is not possible to confidently ascribe these finds to a specific period [29, 54, 65–67].

Finally, another three crucibles related to copper metallurgy, slag residues, two plano-convex moulds with an elongated fissure in the middle, four stone hammers/anvils, a grinding stone/polisher, and two arrowheads were found at Cova Josefina d'Escornalbou (Ruidecanyes), also dated between Chalcolithic-EBA and with unclear stratigraphy [29, 52, 67, 68]. This cave combined habitational traces with human remains of at least four individuals at the bottom of the cave but there were not clear connections between metallurgical finds and these remains [68]. Two flat axes were also reported as potentially coming from the cave, but this is not confirmed [29].

This briefly summary of the evidence shows a very rudimentary, domestic, sporadic, and unspecialised activity carried out in temporary habitats developed within caves (Vapor Gorina is the exception). This differs from the open habitats where metallurgy was conducted in other areas, including southern Iberia and France. Thus, it is important to evaluate if metallurgy was part of a symbolic sphere. Although burials were conducted in some of these caves, there is no compelling evidence to relate metallurgical activities to the deceased, except for Cova del Buldó. However, in later limes (*ca*. 1600 cal. BCE) the metallurgist tomb at Forat de la Tuta (Riner, Lleida) makes this relationship evident [63]. In the remaining sites summarised, the habitational contexts where metallurgy was carried out are not contemporaneous to the bodies inhumated, and it should be noted that habitat in caves is the rule for most of the northeastern territory during Chalcolithic times (see The social value of Chalcolithic copper in the Western Mediterranean). Moreover, level II.5 of La Bauma (see Materials analysed and their archaeological contexts) and the slightly later context at Cova de l'Heura are more indicative of metallurgical production areas (sometimes referred to as workshops in the literature quoted) than ritual environments. Thus, metallurgy cannot be confidently related to an esoteric sphere in this case, as opposed to other cases such as that of early Italian metallurgy, also conducted within caves [69].

When comparing the assemblage from La Bauma (see Chalcolithic metallurgy in the Northeast: between two technological traditions) to other sites described in this section, a quantitative difference can be observed. In the majority of these caves, one to three crucibles were found, contrary to the 65 sherds recovered at La Bauma. At the same time, the scarcity of isotopic information prevents a more detailed comparison of resource acquisition strategies among different sites.

## Materials analysed and their archaeological contexts

Seven small crucible sherds (Fig 2) kept at Museu de la Garrotxa (Olot) were analysed. They all had copper residues attached to their internal surfaces. Importantly, none of these fragments corresponds to those analysed in Alcalde *et al.* [31]. The term 'crucible' is used here as a shorthand to denote all kinds of ceramics used for metallurgical processes, without implying they were purpose-made or specialised vessels unless this is explicitly mentioned. To simplify the naming of the analysed samples, we refer to them in this paper by the square in which they were found. The complete sample references can be found in Table 1.

Before the study of these seven crucibles was conducted, metallographies of another three crucibles from La Bauma had been conducted but never published (see Chalcolithic metallurgy in the Northeast: between two technological traditions). With the aim of making these older results available, a short description of these older micrographs is provided below (see Results) in relation to the newer analyses. The references for these samples are PA6325, PA6326 and PA6327.

G11, F12 and PA6325 show incised decoration. F11 and G12 are part of the same vessel. Although metallography was conducted on F11 and LIA on G12, we refer to the results from these analyses as F11 throughout the text to avoid confusion.

The sherds analysed were found in five different levels that had been previously dated by $^{14}$C analyses on charcoal samples [31]. Results were re-calibrated with the IntCal20 curve (95.4%) using ORAU-OxCal software, but the large error of some samples does not allow to establish a reliable chronology. Moreover, Soriano Llopis [29] has already expressed his doubts about the delimitation of *estratos* (strata, *i.e*. II, III, etc.) and finer *niveles* (layers, *i.e*. II.4, II.5, etc.) by the excavators of La Bauma, raising questions on the potential mixture of materials between layers. This is confirmed by the fit of F11 with G12, which were recovered from

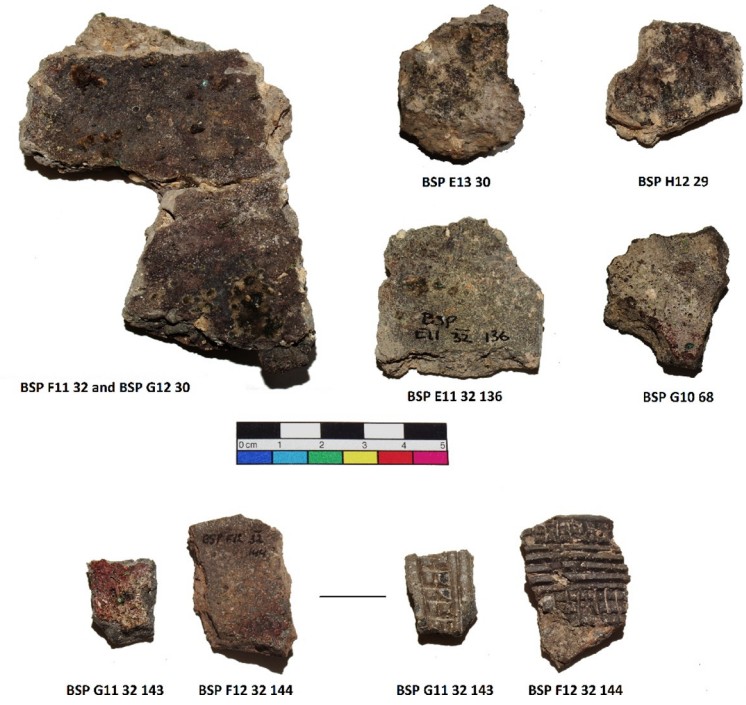

**Fig 2. Crucible sherds analysed.**

different strata (Table 1). However, the assessment of the material culture recovered together with the $^{14}$C dates available, allow us to characterise these contexts within the Chalcolithic Bell Beaker horizon, in the 3$^{rd}$ millennium BCE. The information available for the older, previously unpublished, samples studied by S. Rovira positions them in layers II.4 and II.5, supporting a chronological relationship with the rest of the assemblage analysed.

All contexts have been described in Alcalde *et al.* [31, 70, 71] and Soriano Llopis [29]. Level II.4, considered the most recent context, consisted of a rectangular hut of domestic use that was protected by the cave and built out of perishable materials. It had no pavement. The hearth where metallurgical processes took place was located outside the hut, while other subsistence activities involving the use of grinding stones were located inside. A burial of multiple individuals within a pit was also found in this level. Macroscopic analysis of the crucible sherds recognised three non-decorated sherds and two decorated ones related to this horizon.

Level II.5 has been characterised as a metallurgical production context with no habitational traces. Here, three unusually large hearths with thermally altered rocks in them were discovered. They were located outside the area protected by the cave, which would have facilitated the evacuation of smoke. Around them, residues related to the pyrometallurgical operations were dispersed including the highest amount of crucible sherds: 37, seven of which were decorated.

Finally, level III.1 comprises a domestic setting consisting of a rectangular hut made of perishable materials with a pavement of calcareous rocks. Inside of it, a hearth was found in which the metallurgical operations took place. The material culture found clearly differentiated separate uses of the spaces inside and outside the hut. 20 crucible sherds were found, six of them decorated.

## Methods

Qualitative elemental analysis was conducted on the inner and outer surfaces of 32 sherds, prior to sub-sampling the technical ceramics with higher metal contents. These screening analyses were performed by portable x-ray fluorescence (pXRF) using an Olympus Innov-X Systems Delta Premium and an Innov-X Alpha equipment. In both cases, the so-called Soils mode was activated, which uses a Compton-normalised algorithm and is optimised for the detection of minor and trace elements in silica-rich matrices.

Six sherds were further sampled for microscopic examination, mounted in epoxy resin, and ground and polished flat down to 1μm using SiC disks and diamond paste [72]. They were later documented using a Leica M205 stereomicroscope.

The prepared samples were analysed using compound optical microscopy (Leica DM4000 and Keyence VHX-6000) under plain polarised (PPL) and cross-polarised (XPL) light, prior to carbon-coating and analysis by SEM-EDS (ZEISS GeminiSEM 300, Ultim Max Silicon Drift EDS detector from Oxford Instruments, 20kV, 8.5mm WD, 4.5 spot size, process time 5, 700,000cpa).

EDS spectra were processed using Aztec software. Compositions are given in wt%, reporting only concentrations that were above 3σ of the background. Normalised results are provided but analytical totals are indicated. Analytical totals are usually between 95–105%, except for very porous or corroded areas where totals dropped considerably. Non-metallic phases are reported as conventional compounds with the oxygen calculated by stoichiometry, whereas oxygen in metallic phases is reported as measured.

Bulk compositions of ceramic pastes and slag layers reflect the mean composition of at least three different measurements of a standardised area at x100 magnification (1065x795μm) that included inclusions and porosity if present. In some instances, due to thinness of the slag layers, this area had to be reduced accordingly to acquire representative analyses. The composition of the glassy and ceramic matrices corresponds to the mean of three different analyses in which inclusions and porosity were avoided. In this case, it was not possible to use a standardised area of analysis as samples were highly heterogeneous.

Compositions of other metallic or mineral phases in the slag and ceramic correspond to a single analysis. Area analyses were preferentially used, except for very small phases.

LIA were conducted on the slag of five samples via multi collector-inductively coupled plasma-mass spectrometry (MC-ICP-MS) at the Geochronology and Geochemistry Service (SGIker) at the University of the Basque Country (Spain) using a Neptune (Thermo Fisher Scientific) spectrometer. A Thallium reference material NBS997 with a normalised ratio of 205Tl/203Tl = 2.3889 was used for the internal mass correction. The reliability and reproducibility of the method were verified by regular measurements of the certified reference material NBS981 interspersed between the measurements of the samples, and in the same conditions. Uncertainties of measurements are smaller than symbols used in all graphs. For further methodological details related to LIA see Murillo-Barroso *et al*. [12] and Rodríguez *et al*. [73]. No permits were required for the analysis of geological samples. Permits for the analyses of archaeological samples were granted by the Chief Curator and Head of the Museu de la Garrotxa, under the R&D project: PN623183 funded by the European Commission.

## Results

The following sections summarise our results. Further micrographs and SEM-EDS results can be found in the supporting information (S1 File) attached to this paper.

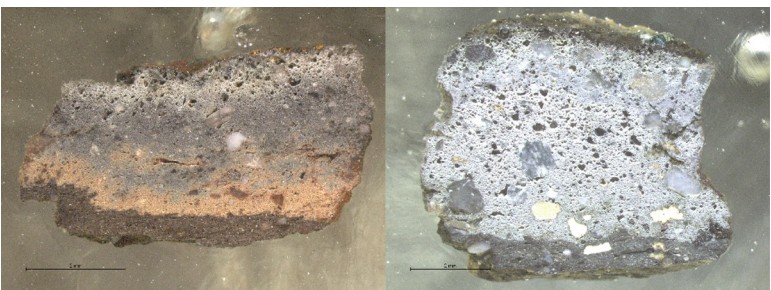

**Fig 3. Samples E13 (left) and F12 (right).** Note the different ceramic paste colours.

### Technical ceramics characterisation

The ceramic pastes of all samples are different macroscopically (Fig 3). Samples have layers of different colours ranging from dark brown and grey (outer part) to white (inner part). Changes in colour are associated to thermal alteration of the pastes and variable redox environments. Large bloating pores are progressively developed towards the inner surfaces what indicates that these crucibles were heated from the inside. All crucibles had a thin layer of slag developed on top of their inner surfaces.

In spite of macroscopic differences in colour, bulk (Table 2) and ceramic matrix analyses (Table 3) indicate use of similar clays for crucible making. Technical ceramics are made of an aluminosilicate (19.1–24.1wt%$Al_2O_3$) mainly containing FeO, $K_2O$ and CaO. MgO, $Na_2O$ and $TiO_2$ are minor components. Sporadic minor impurities of MnO and $P_2O$ can be also related to the clay used.

The ceramics contain different kinds of mineral inclusions (Table 4). These are very poorly sorted (from half millimetre to a few micrometres in diameter) and generally of angular and subangular shapes when not chemically altered. Characterisation of mineral inclusions was not exhaustive, but aimed to provide a general overview of me main types of clay particles. A common trend that includes quartz, different kinds of feldspars, Fe aluminosilicates, and other minerals containing Ti and Zr can be seen. Moreover, samples G10 and H12 present burnt out organic inclusions too (Fig 4).

The matching composition of the pastes and kinds of inclusions is consistent with the purported use of the same clays for all of them. The varied size of the mineral inclusions as well as the ununiform presence of organic particles indicates that the clay probably naturally contained all these materials, so no conscious addition of temper occurred. It might be the case that the alteration of the clay in some crucibles is masking other instances of organic inclusions. It is noted, however, that the mineral characterisation of inclusions was performed by

**Table 2. SEM-EDS results (wt%) of bulk ceramic analyses.**

| ID | $Na_2O$ | MgO | $Al_2O_3$ | $SiO_2$ | $P_2O_5$ | Cl | $K_2O$ | CaO | $TiO_2$ | FeO | Analytical Total |
|-----|------|-----|------|------|--------|--------|------|-----|--------|-----|------------------|
| E13 | 0.4 | 1.1 | 18.0 | 62.6 | <LOD | <LOD | 3.0 | 4.4 | 0.7 | 9.7 | 70.2 |
| F11 | 0.6 | 1.5 | 19.4 | 63.6 | <LOD | <LOD | 3.8 | 2.7 | 0.7 | 7.8 | 71.0 |
| F12 | 1.1 | 1.3 | 20.3 | 61.6 | 1.7 | <LOD | 3.9 | 2.7 | 1.0 | 7.7 | 67.7 |
| G10 | 0.4 | 0.9 | 15.7 | 71.0 | <LOD | 0.1 | 2.8 | 1.1 | 1.2 | 6.9 | 71.9 |
| G11 | 0.8 | 1.2 | 20.1 | 62.9 | 1.1 | <LOD | 3.8 | 1.3 | 0.8 | 8.7 | 60.3 |
| H12 | 0.5 | 1.6 | 20.6 | 59.8 | 0.3 | 0.1 | 4.5 | 3.7 | 0.7 | 8.3 | 59.8 |

<LOD = below limits of detection.

**Table 3. SEM-EDS results (wt%) of ceramic matrix analyses.**

| ID | Na$_2$O | MgO | Al$_2$O$_3$ | SiO$_2$ | P$_2$O$_5$ | Cl | K$_2$O | CaO | TiO$_2$ | MnO | FeO | Analytical Total |
|---|---|---|---|---|---|---|---|---|---|---|---|---|
| E13 | 0.4 | 1.4 | 23.0 | 58.8 | <LOD | <LOD | 3.7 | 2.4 | 0.9 | 0.5 | 9.5 | 96.2 |
| F11 | 0.7 | 1.5 | 23.6 | 63.2 | 1.6 | <LOD | 5.0 | 1.2 | 0.7 | <LOD | 4.7 | 95.8 |
| F12 | 0.6 | 1.7 | 24.1 | 60.0 | <LOD | <LOD | 3.4 | 1.4 | 0.7 | <LOD | 8.2 | 88.8 |
| G10 | 0.4 | 1.1 | 19.1 | 66.1 | <LOD | 0.2 | 3.2 | 1.0 | 0.8 | 0.3 | 8.2 | 72.7 |
| G11 | 1.4 | 1.4 | 22.0 | 61.4 | 0.4 | 0.1 | 7.3 | 1.0 | 0.7 | <LOD | 5.6 | 84.4 |
| H12 | 0.5 | 1.7 | 22.4 | 60.8 | 0.5 | 0.1 | 5.1 | 2.3 | 0.9 | <LOD | 6.5 | 84.2 |

<LOD = below limits of detection.

SEM-EDS and is not fully comprehensive–thin section analyses could consolidate and expand this interpretation by providing better-grounded mineralogical information.

## Characterisation of the metallurgical operations

Focusing on the slag layers, their bulk compositions (Table 5) shows broadly comparable levels of SiO$_2$ (40.5wt%SiO$_2$−53.1wt%SiO$_2$) and Al$_2$O$_3$ (12.5wt%Al$_2$O$_3$−18.6wt%Al$_2$O$_3$), with a SiO$_2$/Al$_2$O$_3$ ratio around 3 and therefore comparable to that in the ceramics proper (Table 6). This suggests that much of the slag may simply be molten ceramic. Notable differences, particularly in the concentrations of MgO, P$_2$O$_5$, CaO and CuO, are corroborated by ratios relative to Al$_2$O$_3$ (Table 6). Glassy matrices analyses corroborate these patterns (Table 7).

The levels of CuO in the slag range from 4.1wt% to 22.4wt% and represent losses of this metal during the metallurgical operations developed inside these crucibles. Additionally, F11 has considerably higher FeO, but this enrichment appears to derive from the decomposition of Fe-rich minerals in the ceramic (see below). Other elements are potentially indicative of the charge. Sporadic As$_2$O$_3$ (G10 and H12) and PbO (E13) in the slag (Table 5) must be related to the impurities of the ore or metal charge. Slag layers in crucibles G10, G11 and F12 are the thickest in the assemblage (Table 8, Fig 5), showing more interaction with the ceramic and large bloating pores. They also form a distinct compositional group, particularly noticeable in the much higher levels of CaO, MgO and P$_2$O$_5$ (Table 5). Slag in H12 is also relatively thick and enriched in CaO, although to a lesser extent. The combined enrichment in CaO and MgO is suggestive of the former inclusion in the charge of minerals rich in these oxides, such as calcite and dolomite. The slag enrichment in P$_2$O$_5$ relative to the ceramic might derive from charcoal ash, or perhaps from other mineral impurities.

The remaining samples (E13 and F11) are much thinner (Table 8, Fig 6) and glassier, and they show a much more moderate slag enrichment in CaO, which appears coupled with an also moderate increase in K$_2$O (Table 5). This combination may be explained by a contribution

**Table 4. Summary of mineral inclusions identified.**

| ID | Mineral inclusions |
|---|---|
| F12 | Quartz, Fe oxide mineral, Fe aluminosilicate, Chromite |
| E13 | Quartz, K-feldspar, Zircon, Fe aluminosilicate, Fe-Ti aluminosilicate |
| F11 | K-feldspar, Na-feldspar, Rutile, Pb silicate, REE mineral, Zircon, Ilmenite |
| G10 | Quartz, K-feldspar, Fe aluminosilicate, Ilmenite, Zircon |
| G11 | Quartz, K-feldspar, Ilmenite |
| H12 | Quartz, K-feldspar, Na-feldspar, Fe aluminosilicate, Rutile, Zircon |

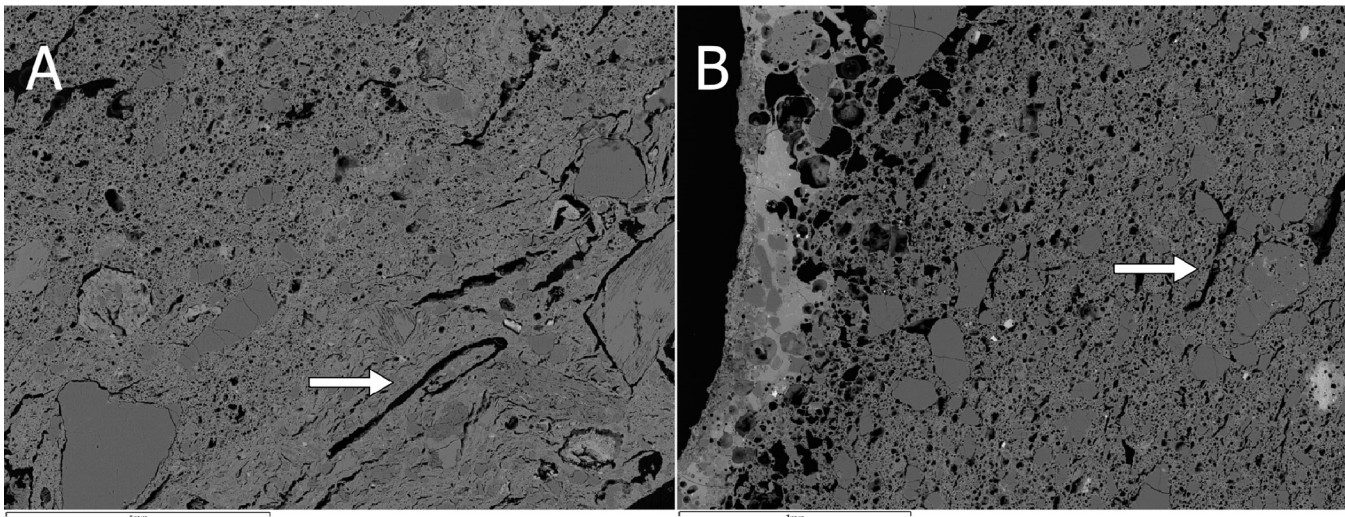

**Fig 4.** BSE micrograph of H12 (A) and G10 (B) with evidence of burnt out organic inclusions in the ceramic paste. (Scales = 1mm).

from charcoal ash rather than geological material, and in any case, it indicates the possibility of a different use.

**Metallurgical operations with Ca-rich charges.** We propose that F12, G10, G11 and H12 are evidence for the possible processing of a copper ore with Ca-rich gangue. Although the slag layers of these samples are very heterogeneous, they share some characteristics.

In addition to the bulk enrichment in Ca and Mg, other aspects of the microanalysis corroborate the former presence of calcite/dolomite in the charge, particularly the presence of neo-silicates containing CaO and/or MgO. In F12, neo-silicates are rich in FeO, CaO and $Al_2O_3$, often with MgO (up to 9.1wt%MgO). In G11, neo-silicates rich in CaO (up to 11.7wt% CaO) appear in clustered formations (Fig 7), which are likely pseudomorphs of semi-dissolved mineral grains. G10 has a glassy matrix populated by anorthite needles and tabular pyroxenes that combine CaO (24.8–25.3wt%CaO), MgO (6.6–8.3wt%MgO) and FeO (up to 10.74wt% FeO). In H12, while the bulk CaO in the glassy matrix is more moderate, there are areas with locally higher concentrations (up to 12.4wt%CaO), particularly located in well-defined clusters of cuprite prills. These clusters could be interpreted as copper ore relicts with Ca-rich gangue (Fig 8A). Moreover, sporadic anorthite crystals are distributed in its glassy matrix. These anorthite crystals are sometimes associated to clusters of Cu-based prills that perhaps might point to ore relicts (Fig 8B).

**Table 5. SEM-EDS results (wt%) of bulk slag analyses.**

| ID | Na₂O | MgO | Al₂O₃ | SiO₂ | P₂O₅ | Cl | K₂O | CaO | TiO₂ | MnO | FeO | CuO | As₂O₃ | PbO | Analytical Total |
|-----|------|-----|-------|------|------|------|-----|------|------|------|------|------|-------|------|------------------|
| E13 | <LOD | 1.1 | 15.4 | 53.1 | 0.6 | <LOD | 4.1 | 3.3 | 0.6 | 0.4 | 6.7 | 14.7 | <LOD | 0.8 | 84.4 |
| F11 | 1.2 | 1.9 | 18.6 | 51.6 | 0.4 | <LOD | 4.4 | 3.3 | 0.6 | <LOD | 14.2 | 6.7 | <LOD | <LOD | 84.8 |
| F12 | 1.2 | 4.3 | 13.6 | 40.5 | 3.6 | <LOD | 3.2 | 22.4 | 0.6 | 0.2 | 6.3 | 4.1 | <LOD | <LOD | 91.7 |
| G10 | 0.5 | 2.4 | 13.3 | 42.8 | 2.8 | <LOD | 3.7 | 19.0 | 0.8 | 0.2 | 5.6 | 8.5 | 2.1 | <LOD | 86.0 |
| G11 | <LOD | 2.3 | 12.5 | 41.8 | 1.5 | <LOD | 2.0 | 12.6 | 0.5 | <LOD | 4.5 | 22.4 | <LOD | <LOD | 92.5 |
| H12 | <LOD | 1.5 | 16.0 | 52.0 | 0.4 | 3.5 | 4.4 | 6.2 | 0.5 | <LOD | 5.0 | 14.5 | 1.0 | <LOD | 82.5 |

<LOD = below limits of detection.

**Table 6. Ratios between different elements of the ceramic and slag bulk compositions of all samples.**

| ID | $Al_2O_3/SiO_2$ | | $MgO/Al_2O_3$ | | $P_2O_5/Al_2O_3$ | | $CaO/Al_2O_3$ | |
|---|---|---|---|---|---|---|---|---|
| | Ceramic | Slag | Ceramic | Slag | Ceramic | Slag | Ceramic | Slag |
| E13 | 3.5 | 3.4 | 0.06 | 0.07 | 0.03 | 0.04 | 0.24 | 0.21 |
| F11 | 3.3 | 2.8 | 0.08 | 0.10 | 0.03 | 0.02 | 0.14 | 0.18 |
| F12 | 3.0 | 3.0 | 0.06 | 0.32 | 0.08 | 0.26 | 0.13 | 1.65 |
| G10 | 4.5 | 3.2 | 0.06 | 0.18 | 0.03 | 0.21 | 0.07 | 1.43 |
| G11 | 3.1 | 3.3 | 0.06 | 0.18 | 0.05 | 0.12 | 0.06 | 1.01 |
| H12 | 2.9 | 3.3 | 0.08 | 0.09 | 0.01 | 0.03 | 0.18 | 0.39 |

All results in wt%.

Delafossite crystals were identified in all the slag samples in this group. Delafossite needles are typically taken as diagnostic of smelting operations in mildly reducing environments, but they can also appear in melting slag [11, 74, 75]. In this case, we are inclined to interpret these as related to smelting, especially when they appear in clusters that might be indicative of ore relicts (Figs 7 and 9). Big clusters of skeletal cuprite next to clusters of Cu-based prills in G11 (Figs 7 and 10) might also indicate the formation of new copper, supporting a smelting process. Presence of dendritic cuprite indicates reaching temperatures above 1232°C, in accordance with crucible-based metallurgy.

In addition to CaO and MgO, the high levels of FeO in some neo-silicates should be discussed. In sample F12, individual crystals contain up to 37.5wt%FeO. In H12, some bright clusters of Fe silicates can be identified (Fig 11). However, given that no copper is associated to these clusters and that similar mineral inclusions are recognised in the ceramic paste of this vessel (Table 4), it is plausible that these slag phases and the overall FeO enrichment derive, at least in part, from semi-dissolved ceramic inclusions. Thus, it is difficult to confidently discern the extent to which Fe was part of the crucible charge. Fayalite and wüstite phases were not found in any sample, in agreement with other early smelting evidence from Iberia [11, 75].

Cu-based prills in metallic or oxide (cuprite) form and variable sizes (from less than 1μm up to 13.3μm of diameter) were trapped in the glassy matrix of all samples in this group (Fig 12). Analyses on F12 show that all of these are nominally pure Cu with no relevant minor elements. Conversely, most prills in G11 have some Fe (typically <1wt%Fe) with a few reaching values as high as 7.4wt%Fe. Moreover, some prills in G11 present minor As (<0.6wt%As) and Ag (<0.6wt%Ag). A single prill was found to contain 0.4wt%As and 0.6wt%Sn. Interestingly, no prill combines the three impurities (As, Ag and Sn), although all of them have some Fe. However, it is necessary to note that the results reported for these minor elements are close to the detection limits of the SEM-EDS (~0.1–0.3wt%).

**Table 7. SEM-EDS results (wt%) of glassy matrix analyses.**

| ID | $Na_2O$ | MgO | $Al_2O_3$ | $SiO_2$ | $P_2O_5$ | Cl | $K_2O$ | CaO | $TiO_2$ | MnO | FeO | CuO | $As_2O_3$ | PbO | Analytical Total |
|---|---|---|---|---|---|---|---|---|---|---|---|---|---|---|---|
| E13 | 0.6 | 1.2 | 16.9 | 60.5 | 0.8 | <LOD | 5.0 | 4.6 | 0.6 | <LOD | 5.1 | 4.7 | <LOD | 0.8 | 99.3 |
| F11 | 0.9 | 1.3 | 19.7 | 51.9 | 0.5 | 0.2 | 3.9 | 3.8 | 1.0 | <LOD | 16.3 | 1.4 | <LOD | <LOD | 96.0 |
| F12 | 1.9 | 4.7 | 13.6 | 41.9 | 2.1 | <LOD | 2.1 | 28.1 | 0.5 | 0.3 | 3.3 | 2.0 | <LOD | <LOD | 99.3 |
| G10 | 0.6 | 2.0 | 13.5 | 47.7 | 2.0 | <LOD | 4.3 | 23.3 | 0.8 | <LOD | 5.4 | 3.8 | 1.4 | <LOD | 103.0 |
| G11 | <LOD | 2.8 | 13.9 | 43.7 | 1.9 | <LOD | 2.3 | 15.5 | 0.6 | <LOD | 6.1 | 13.3 | <LOD | <LOD | 98.5 |
| H12 | 0.5 | 1.8 | 18.6 | 49.4 | 0.4 | <LOD | 4.4 | 6.2 | 0.7 | <LOD | 6.4 | 11.7 | 1.3 | <LOD | 93.3 |

<LOD = below limits of detection.

**Table 8. Thicknesses of the slag layers.**

| ID | Min. slag thickness (μm) | Max. slag thickness (μm) |
| --- | --- | --- |
| E13 | 125 | 375 |
| F11 | 35 | 119 |
| F12 | 234 | 547 |
| G10 | 131 | 789 |
| G11 | 400 | 1833 |
| H12 | 58 | 500 |

Metallic and cuprite prills in G10 also contain Fe (0.8–1.4wt%Fe). In all cases, they are either pure Cu or Cu with minor amounts of As (0.7–2.0wt%As). Finally, H12 presents metallic prills that are generally pure Cu, although in one case As (1.8wt%As) and Fe (8.2wt%Fe) were recorded.

Overall, this evidence indicates that these are smelting crucibles where copper ores with Ca-rich gangue were processed. Fe and As are also part of the minor components of this ore, as well as sporadic Sn and Ag in very small amounts.

As mentioned above (see Chalcolithic metallurgy in the Northeast: between two technological traditions), in the first study of materials from La Bauma, metallographic analyses were carried out on three crucible fragments. Preliminary chemical analyses of the surface had identified characteristic copper peaks in all of them [31]. With the knowledge now gained about the structure and composition of the slags from this site, it is possible to make a more precise interpretation of some selected images.

Sample PA6325 corresponds to the upper part of the crucible, including its rim. The thickness of the slag layer is above 200μm. Fig 13 shows the slag matrix with cuprite and its exsolution, what can be interpreted as a relic of the processed ore. In the vitreous matrix, there are also crystallisations of a probable neo-silicate.

The slag layer of sample PA6326 is more than 200μm thick in the observed area. It consists of an apparently uniform glassy matrix in which numerous metallic prills of very different sizes are embedded (Fig 14). Finally, the slag in sample PA6327 is similar to the previous one, with abundant metallic prills embedded in the matrix, but in this case, scattered needle formations of a material segregated in the matrix, probably a silicate, are observed (Fig 15). Signs of bloating represented by the bubbles are evident. The thickness of this slag layer is more than 500μm.

**Metallurgical operations with Ca-poor charges.** Turning to the thinner slag in crucibles E13 and F11, they do not present evidence of Ca-rich gangue, and they do not yield conclusive evidence for smelting.

The crystal phases present in E13 are secondary Fe aluminosilicates, anorthite, and delafossite needles (Fig 16). All of them are uniformly distributed, never forming clusters, and therefore cannot be identified as ore relics. However, they are consistent with smelting operations. Given that the mean composition of the glassy matrix reflects minor PbO (0.8wt%PbO), it is likely that it was part of the charge too.

The slag in F11 is particularly thin (Table 8), but it shows the highest FeO concentration in its glassy matrix (16.3wt%FeO). Fe contributions could come from the charge proper during a potential melting process of Cu rich in Fe. However, in some areas, Fe aluminosilicates (similar to those also found in E13) are concentrated in clusters of newly formed crystals (Fig 17). As in E13 above, these were most likely absorbed from the ceramic matrix into this very thin slag, presenting another explanation for Fe enrichment of the glassy matrix.

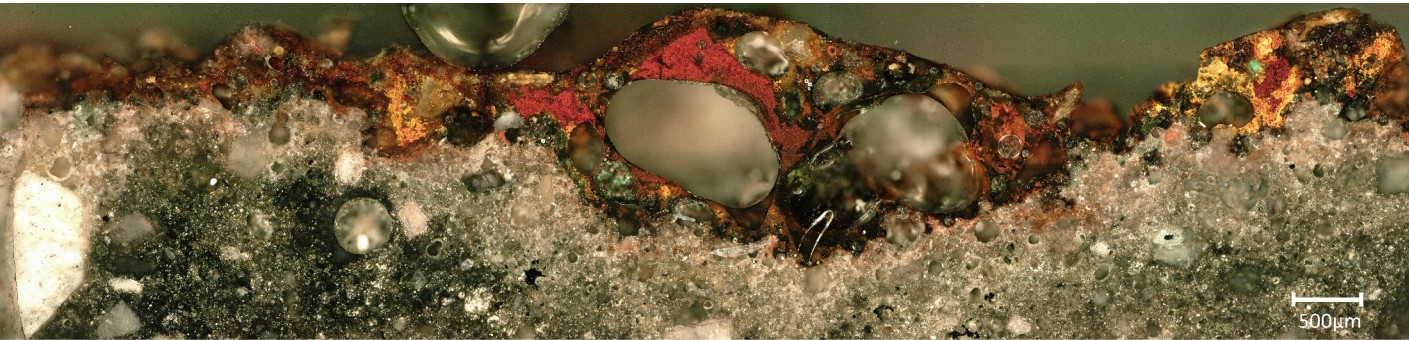

**Fig 5. XPL micrograph of G11 with the thickest of all slag layers analysed.** Note the interaction between the slag layer and the ceramic and the development of bloating pores. (Scale = 500μm).

Cu-rich prills are present in both samples. In E13 all prills are pure Cu with minor amounts of Fe (<2wt%Fe) and only in one case 0.8wt%As was detected. Bright red cuprite exsolutions around metallic Cu clusters are also identified, which could be the result of a re-oxidation process due to the variable redox conditions (Fig 16). In F11, prills are not metallic, but Cu silicates with $As_2O_3$ ranging from 3.3wt%$As_2O_3$ to 8.5wt%$As_2O_3$ and FeO below 1.7wt%FeO.

Overall, no clear copper ore relicts could be identified based on the distribution of mineral phases or microanalytical results. While the presence of copper silicates in F11 is suggestive of a smelting operation, the evidence cannot be taken to conclusively indicate whether smelting or melting took place in these vessels. If smelting did take place, the relatively low CaO and MgO in the slag, together with the presence of PbO in E13 and the different impurities in the Cu prills would strongly suggest the use of different minerals compared to the previous group.

## Lead isotopes

Lead isotopes results of the slag from the five crucibles analysed do not show radiogenic values; all of them were between 18.52–18.63 on the $^{206}Pb/^{204}Pb$ ratio (Table 9). Samples E13 and E11 present quite similar values that might correlate with a common deposit. Results for the remaining three samples (H12, G10 and F11, Fig 18) slightly differ from this first group, what supports the exploitation of different copper resources. Interestingly, samples from La Bauma follow the same trend than previous northeastern samples analysed, filling an existing gap (Fig 18), perhaps indicating that all minerals used originated in this region.

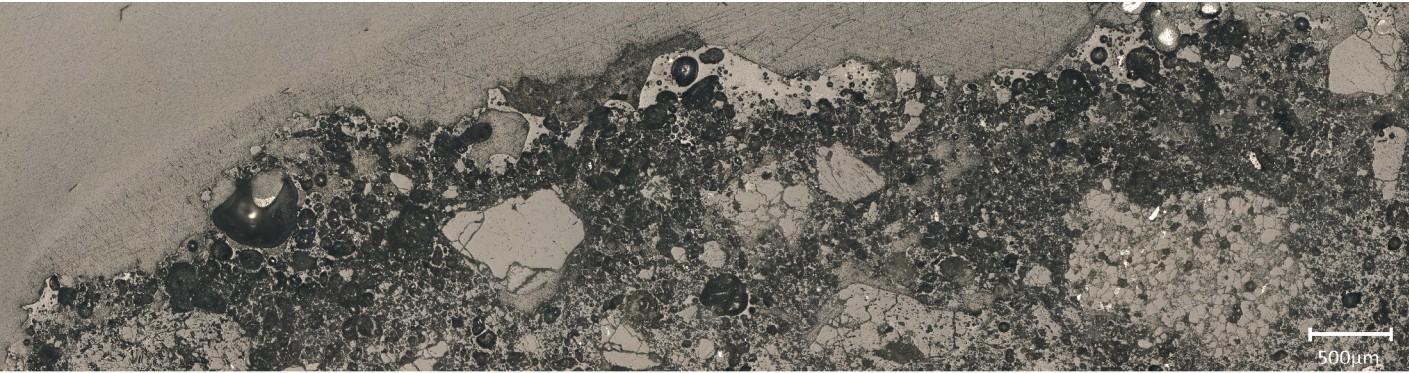

**Fig 6. PPL micrograph of E13 with a very thin slag layer.** Note the great interaction between the ceramic paste and the slag layer and the development of bloating pores. (Scale = 500μm).

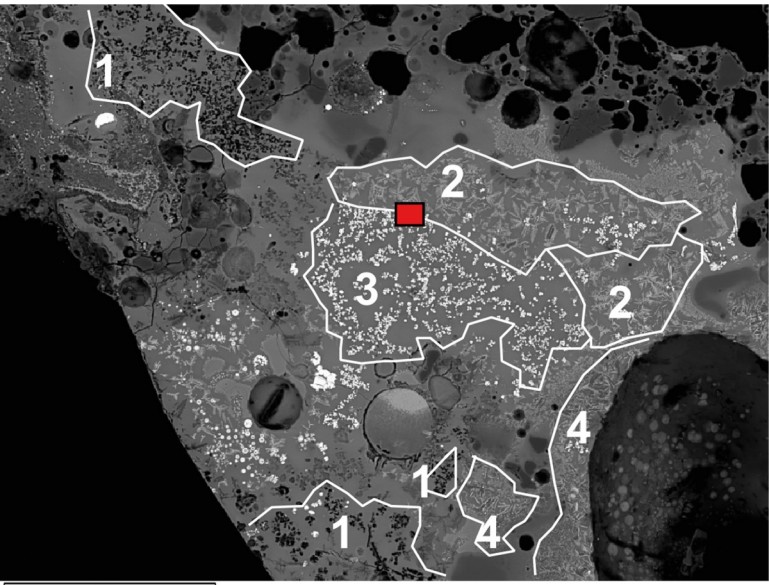

**Fig 7. BSE micrograph of G11 showing clusters of Ca-rich minerals (1), cuprite skeletal formations (2), metallic copper (3) and delafossite(4).** The red square shows the image zoomed-in on Fig 10. (Scale = 240μm).

Regarding provenance and considering that we are dealing with slag layers here, one should consider the potential lead contribution from the three main sources of slag formation: *i.e.* the ceramic paste, the copper ore and the fuel. Rademakers *et al.* [76] have shown that crucibles made of Pb-rich silt (20–1500ppm Pb in the ceramic paste) can affect lead isotopic compositions of their slags layers. Nonetheless, this contamination is expected to be minor when Pb-poor clay–or fuel–is used and especially if Pb-rich ores are smelted. For the case of La Bauma, most of the copper ores known in the Northeast and considered as potential sources here

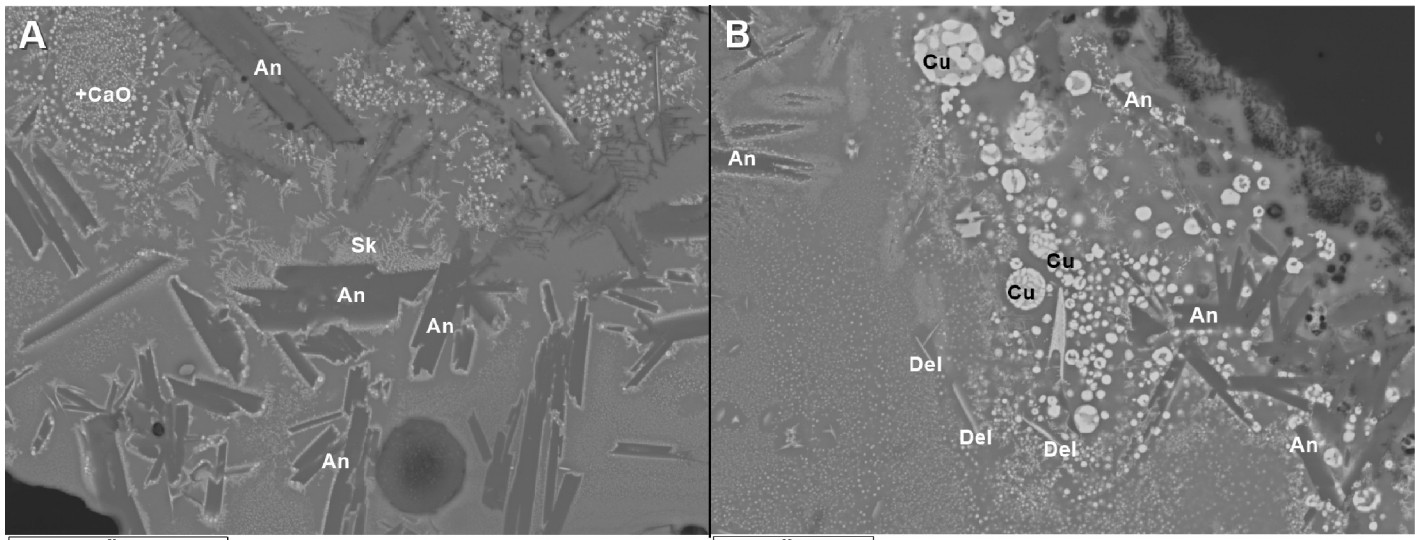

**Fig 8.** A) BSE micrograph of H12 showing anorthite crystals (An) in a glassy matrix with cuprite exsolutions and skeletal formations (Sk). Note the rounded cluster of cuprite prills at the top left that has higher concentrations of CaO. B) BSE micrograph of H12 showing a cluster of Cu-based prills (Cu) with anorthite needles (An) and some delafossite (Del) crystals. (Scales = 50 and 25μm respectively).

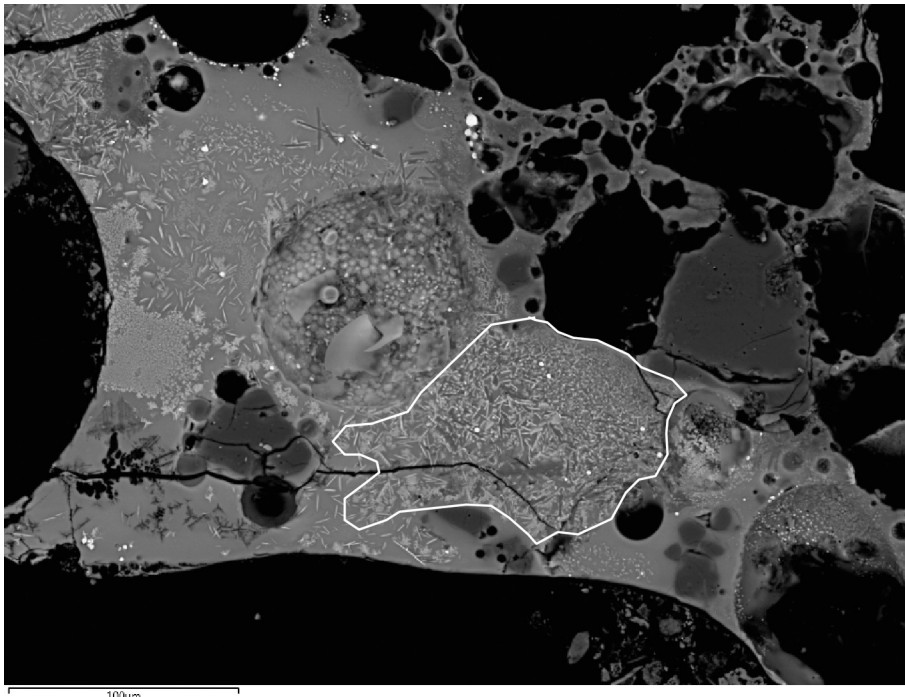

**Fig 9. BSE micrograph of G11 showing a cluster of delafossite needles in the middle of the sample.** It is indicated with a white line. (Scale = 100µm).

(except for Turquesa mine) have significant levels of lead, some of them being actually Cu-Pb minerals [54, 58]. Therefore, despite lacking trace element composition of the ceramic pastes or surrounding clays, it is assumed that the main lead contribution to the slag will be that of

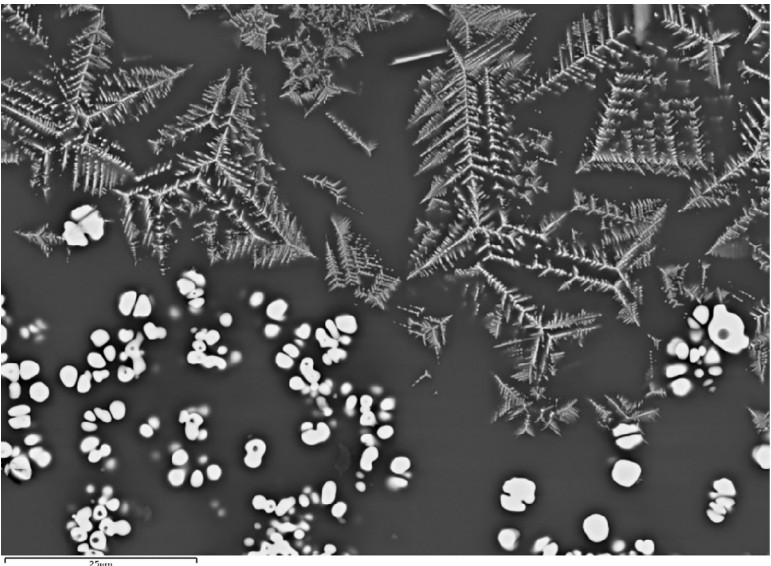

**Fig 10. BSE micrograph of G11 showing a closer view of part of Fig 7.** Metallic Cu-based prills can be seen together with skeletal cuprite. (Scale = 25µm).

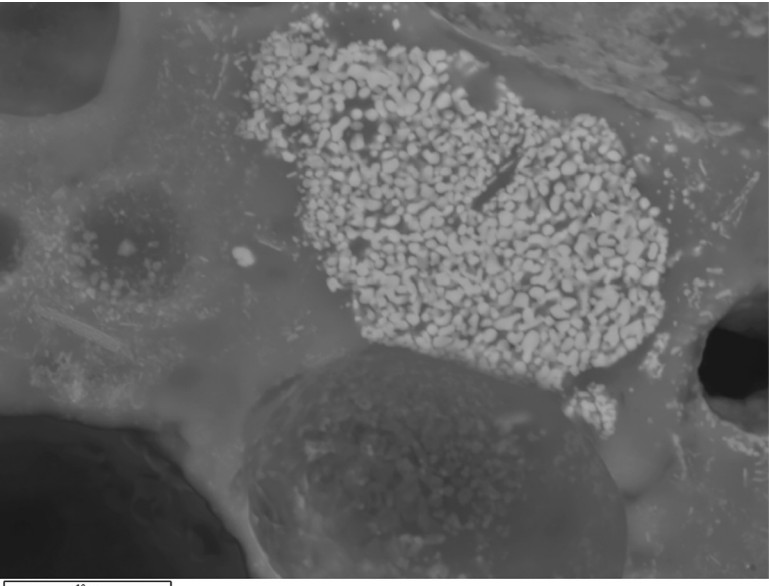

**Fig 11. BSE micrograph of H12 showing a partially dissolved Fe silicate mineral in the glassy matrix.** Note it is not surrounded by Cu-bearing phases. (Scale = 10μm).

the copper ore, and therefore it would be possible to match La Bauma samples with the regional mining districts.

The well-known Molar-Bellmunt-Falset (MBF) mining district (Tarragona province) must be discarded, as well as most of the Catalan Coastal Range isotopically characterised so far, as the available data differ from the archaeological samples (values <2.095 in $^{208}Pb/^{206}Pb$ or <18.4 in $^{206}Pb/^{204}Pb$). The closest concordance for two samples (H12 and G10) is observed

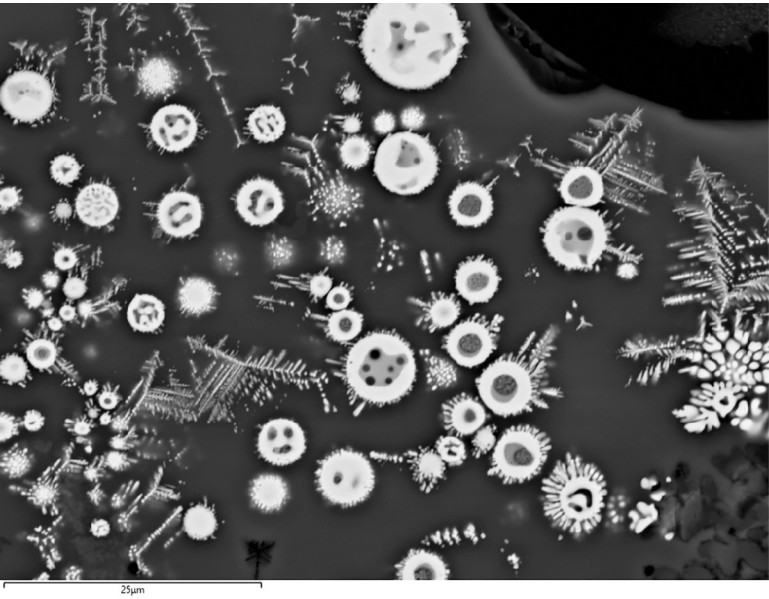

**Fig 12. BSE micrograph of G11 showing numerous Cu-based prills of various sizes and some skeletal cuprite embedded in the slag.** (Scale = 25μm).

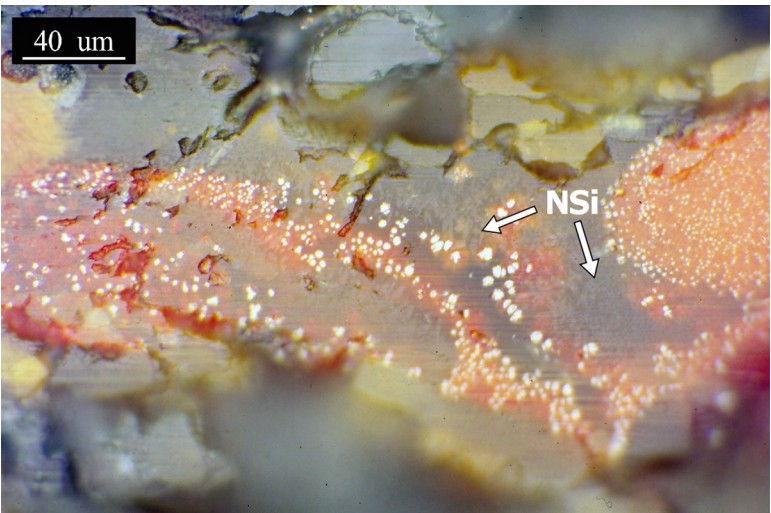

**Fig 13. PPL micrograph of PA6325 showing whitish prills of cuprite, its reddish exsolution and small neosilicates (NSi, light grey) in the darker grey matrix.** (Scale = 40μm).

with the Montsant area (Tarragona province) (Fig 19). Remarkably, Solana del Bepo mine, that is within this area, has highly similar values to H12 in all scatterplots of Fig 19. The extensive study of this mine [42] has evidenced its exploitation in prehistoric times. Other samples from Cova de l'Heura and possibly Vapor Gorina (see Evidence of crucible metallurgy in the Northeast: Bauma del Serrat del Pont and its contemporaneous assemblages) have also been linked to Solana del Bepo mine, although the fit of H12 within this isotopic field is clearer.

G10 has a quite similar isotopic signature than an analysed sample from Balma del Duc. This suggests use of a common local source. Turquesa mine, also within the Montsant area, has been already proposed as the possible source for the copper found in the crucible from Balma del Duc (see Evidence of crucible metallurgy in the Northeast: Bauma del Serrat del Pont and its contemporaneous assemblages) [54, 57, 58], and prehistoric mining has been

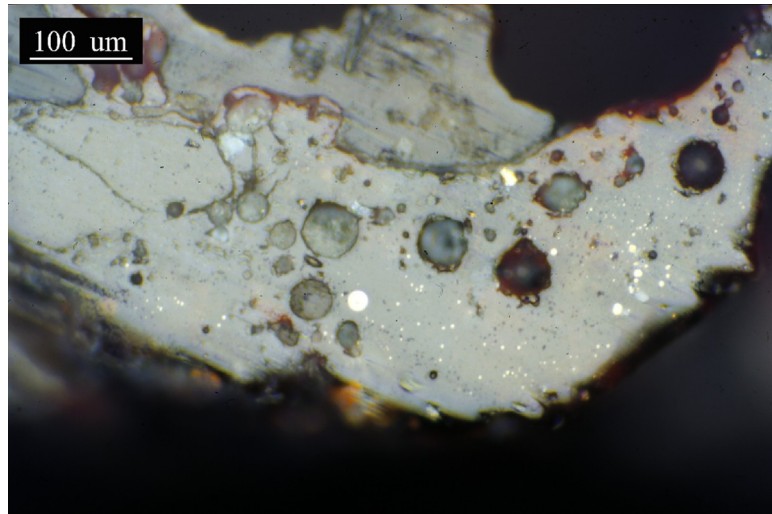

**Fig 14. PPL micrograph of PA6326. Grey glassy matrix embedding numerous metallic prills.** In the upper part of the image, large vacuoles can be seen resulting from the bloating of the ceramic. (Scale = 100μm).

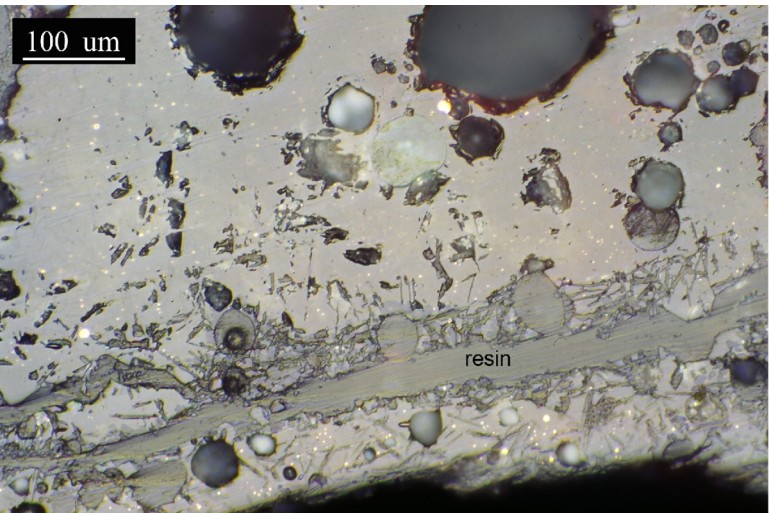

**Fig 15. PPL micrograph of PA6327. Grey glassy slag matrix with abundant metallic prills.** Silicate needles (dark grey) are scattered throughout the matrix. (Scale = 100μm).

documented there. However, differences in the $^{207}Pb/^{204}Pb$ ratio between G10 and the sample from Balma del Duc and Turquesa mine (15.668 *vs.* 15.687) do not allow to confidently ascribe G10 to this mine. Detection of arsenic in G10 would be more consistent with Turquesa mine than with Solana del Bepo mine, as these two mines differ in the presence and absence of arsenic respectively. Whatever the case, both mines are at *ca.* 200km away from La Bauma.

These two samples (H12 and G10) were previously identified as Ca-rich (see Characterisation of the metallurgical operations), which would be consistent with both Turquesa and Solana del Bepo mines. Crandallite ($CaAl_3[PO_4][PO_3OH][OH]_6$) is the most common mineral in the superficial zone of the deposit at Turquesa mine, and there are also small quantities of fluorapatite ($Ca_5[PO_4]_3F$) [80]. Solana del Bepo is a Ba-Cu vein deposit which may be accompanied by variable quantities of quartz, calcite and siderite. Its geological context is largely

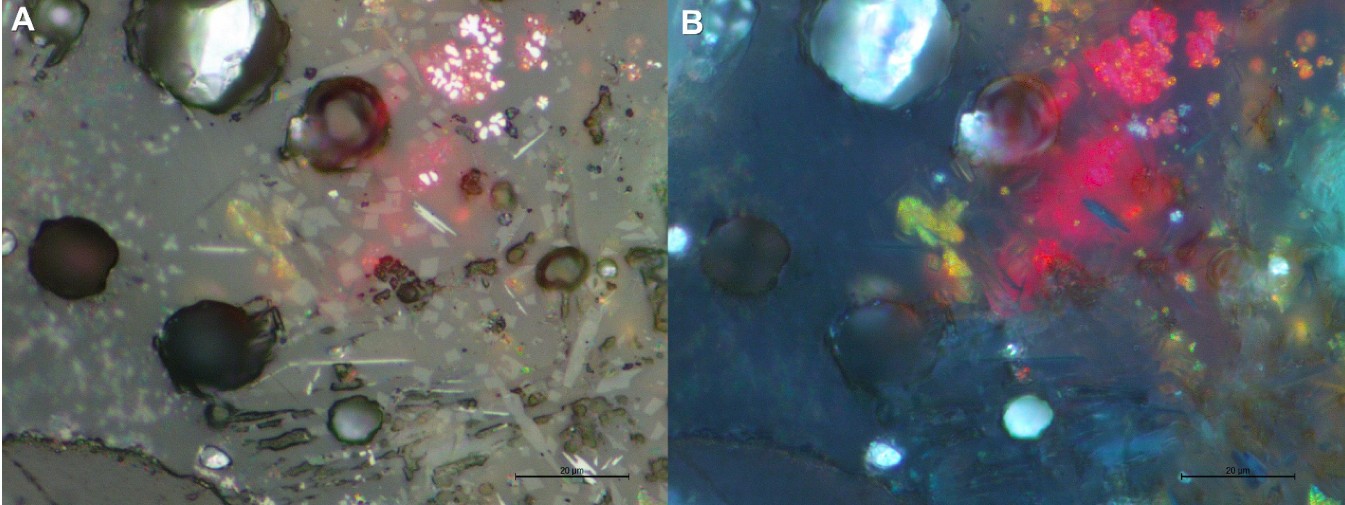

**Fig 16.** PPL (A) and XPL (B) micrographs of E13 showing a bright metallic Cu cluster (top right, Fig 13A) being re-oxidised into red cuprite (top right, Fig 13B). On Fig 13A, white needles were identified as delafossite, light grey crystals as Fe aluminosilicates, and dark grey crystals as anorthite. (Scales = 20μm).

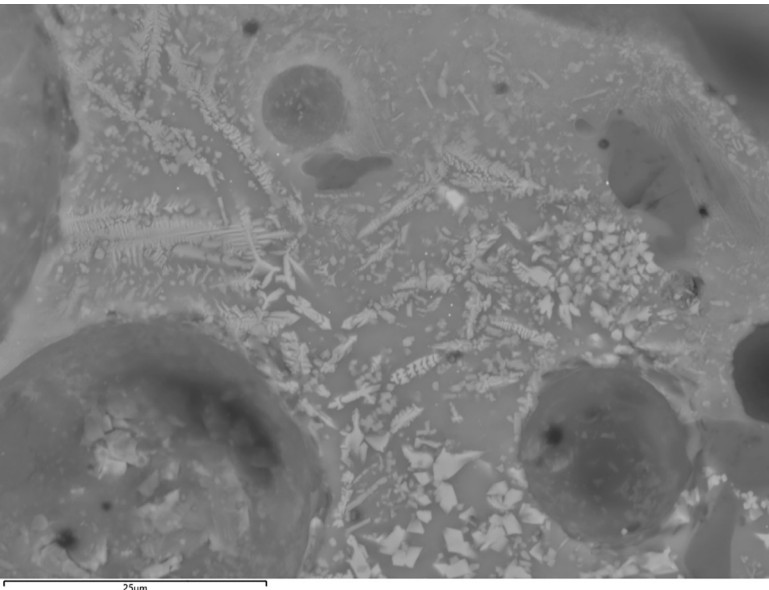

**Fig 17. BSE micrograph of F11 showing different crystal formations of Fe aluminosilicates in the glassy matrix.**
Note that there are no metallic phases around. These crystals are located in very specific areas of the slag.
(Scale = 25μm).

made up of slates affected to a greater or lesser extent by contact metamorphism from the Carboniferous-Permian, above which, locally, there are limestones of the Cornudella Group from the Palaeocene-Eocene [81]. Therefore, it would not be surprising that some calcium rich mineral entered the charge with the copper ores.

The possible copper origin of F11 could be at Les Ferreres mining district, located at the eastern Pyrenees (Fig 1). It is only *ca*. 18km way from La Bauma and it has been already proposed as a possible source of polymetallic Cu-Sn ores also used at the site (see Chalcolithic metallurgy in the Northeast: between two technological traditions). Although this sample is slightly distant in the $^{207}Pb/^{204}Pb$ ratio (Fig 19), this area should be kept in mind as a possible source as only four copper ore samples have been used to characterise this outcrop. This sample was previously identified as Ca-poor (see Characterisation of the metallurgical operations). That would be consistent with Les Ferreres mining district, surrounded by granite, and therefore with less probability of calcium contaminating the crucible charge. This area was also proposed as the copper source of an awl from Cap de Barbaria II (*ca*. 1662–1490 cal. BCE) in

**Table 9. LIA results for La Bauma and Les Ferreres samples (MC-ICP-MS).**

| Site | Type | ID | 208/206 | 207/206 | 206/204 | 207/204 | 208/204 |
|---|---|---|---|---|---|---|---|
| BSP | Slag | E13 | 2.08509 | 0.84364 | 18.6293 | 15.7165 | 38.8440 |
| BSP | Slag | H12 | 2.09332 | 0.84713 | 18.5246 | 15.6928 | 38.7780 |
| BSP | Slag | F11 | 2.09062 | 0.84603 | 18.5812 | 15.7204 | 38.8464 |
| BSP | Slag | E11 | 2.08377 | 0.84305 | 18.6368 | 15.7119 | 38.8349 |
| BSP | Slag | G10 | 2.08595 | 0.84549 | 18.5314 | 15.6683 | 38.6558 |
| Les Ferreres | Cu ore | PA21878 | 2.08913 | 0.84543 | 18.5839 | 15.7114 | 38.8241 |
| Les Ferreres | Cu ore | PA21879 | 2.08912 | 0.84552 | 18.5807 | 15.7103 | 38.8173 |
| Les Ferreres | Cu ore | ROC-1 | 2.09013 | 0.84657 | 18.5547 | 15.7080 | 38.7819 |
| Les Ferreres | Cu ore | ROC-3 | 2.09001 | 0.84656 | 18.5558 | 15.7086 | 38.7820 |

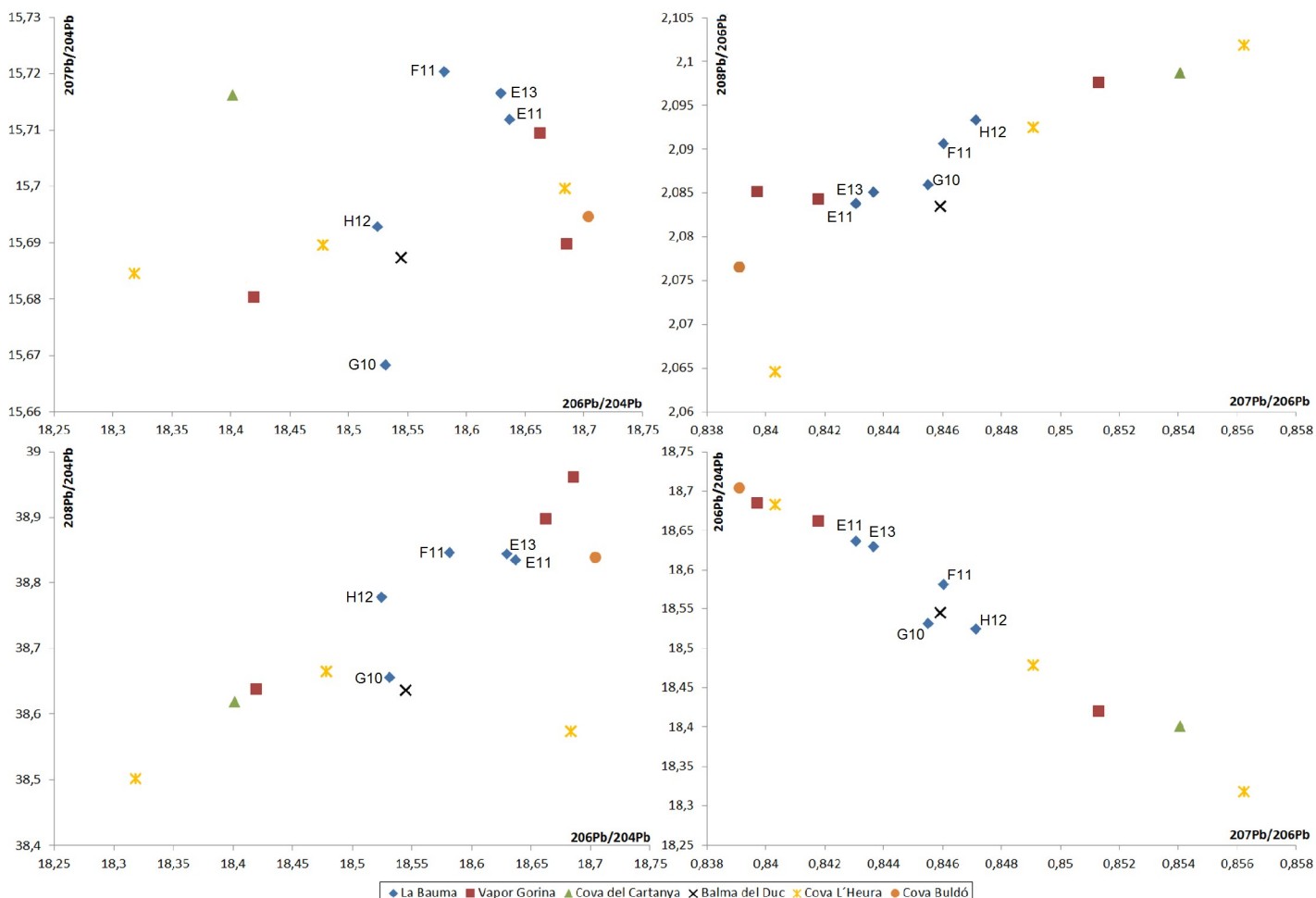

**Fig 18. LIA results for La Bauma and other sites located in the Northeast.**

Formentera island [82], what would evidence exploitation of this mine over an extended period of time. Further analyses in this mining district would be beneficial for future research. In the same isotopic area, only the Palmela point from the Bell Beaker site of Fortin du Saut (Châteauneuf-lès-Martigues, Bouches-du-Rhône, Southern France) was found, although this sample also differs in the $^{207}$Pb/$^{204}$Pb ratio [79]. Some evidence of metallurgical production has been also found in this site and LIA from one prill could match the MBF isotopic field in which copper ores with arsenic, antimony and silver are described from Linda Mariquita mine (El Molar) [64].

Finally, the two remaining samples (E13 and E11) differ from the known copper resources of the Northeast, the Pyrenees and southern France. Although no specific provenance can be proposed with the available data, these samples follow the same trend observed for all the northeastern metallurgical debris analysed so far. It seems possible that some regional resources (but still different ones than the characterised up to now) could have been used in these cases as well, as there are still some lacunas of isotopic information related to some mining districts in the area.

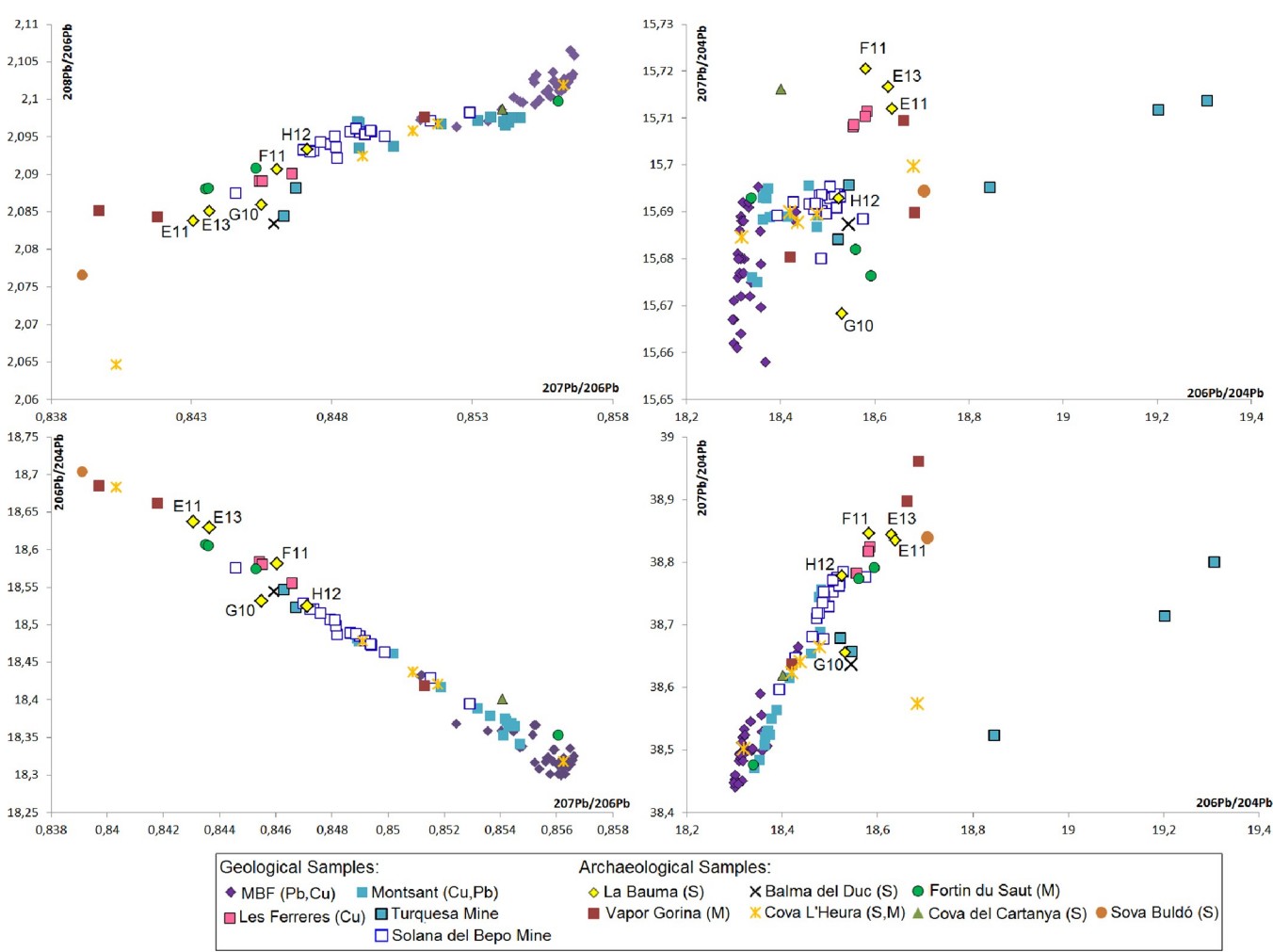

**Fig 19. LIA results for samples from La Bauma, other sites located in the Northeast and geological samples of the region.** These include Les Ferreres mine (this paper) and the MBF [77] and Montsant mining districts [54, 58, 78]. Solana del Bepo and Turquesa mines are part of the latter. Not all samples from Turquesa mine (n. 12) are represented in the graphs, as samples from this source have a radiogenic trend that reaches values of 21.750 $^{206}Pb/^{204}Pb$; 15.857 $^{207}Pb/^{204}Pb$; 38.898 $^{208}Pb/^{204}Pb$; 0.7291 $^{207}Pb/^{206}Pb$ and 1.788 $^{208}Pb/^{206}Pb$. In archaeological samples, S = Slag; M = Metals. Data for archaeological samples was taken from Montero Ruiz [58] and Laubane [79].

## Discussion and conclusions

Analyses of the metallurgical assemblage from La Bauma confirm the use of four vessels (F12, G10, G11 and H12) for smelting of copper ores with a gangue that was rich in CaO, MgO and $P_2O_5$. This ore also contains some Fe and As, as well as sporadic Sn and Ag. LI analyses of two samples of this group (G10 and H12) support a provenance from Montsant area, with G10 more likely related to Turquesa mine and H12 to Solana del Bepo mine. LIA were not conducted on F12 and G11.

E13 and F11 slag layers differ compositionally and microstructurally from the other four samples, indicating a different charge. They are also thinner. While not conclusive, the reported evidence might be consistent with smelting rather than melting. If so, this would have involved pure copper ores with minor As, Fe and Pb. When looking at the LIA results, F11 might be related to Les Ferreres mining district, while E13, together with E11 (for which metallographic data is not available) remain unassigned. However, this unknown copper

source is probably within the region, as the analysed samples from La Bauma follow the same trend than previously analysed samples from the Northeast (Fig 18, see Lead isotopes).

Thus, SEM-EDS and LIA data present a consistent set of results. According to LIA, at least four kinds of resources were possibly used at La Bauma: the nearby (*ca.* 18km in a straight line) copper ores from Les Ferreres at the Pyrenees, the distant (*ca.* 200km) copper ores from Montsant mining district (including copper from Solana del Bepo and Turquesa mines) and a fourth source still to be identified. A potential fifth copper ore used could be that of polymetallic nature (Cu-Sn) reported by Alcalde *et al.* [31] that they located, based on compositional analyses, within Les Ferreres mining district too (possibly either from Les Ferreres or Can Manera mines) (see Chalcolithic metallurgy in the Northeast: between two technological traditions).

The analyses of the crucible pastes show that La Bauma crucibles were manufactured using the same clay. If undecorated vessels were purpose-made vessels manufactured for metallurgical operations (still a not conclusive question), these were made in the same way as ordinary decorated pots (G11 and F12) that were repurposed for metallurgical purposes. A parallel phenomenon can be found in the Bell Beaker assemblage from El Ventorro (Madrid), where undecorated and decorated crucibles have very similar pastes between each other and compared to other non-metallurgical vessels [32, 33].

Ceramic pastes from La Bauma combine varied poorly sorted mineral inclusions (Table 4) as well as some organic burnt out materials. The paste was refractory enough to hold until the end of the operation, which was probably conducted at temperatures in excess of 1100˚C (*i.e.* above the melting point of copper). By then, it was close to chemical and thermal collapse, as indicated by the bloating pores and the slag layers consistently enriched by melted ceramic. Presence of mineral inclusions of varied size and of low melting point (such as the Fe minerals dissolved in some of the slag), suggests that no specific treatment of the clay or conscious selection of tempering material was conducted.

These results allow to technologically classify La Bauma metallurgy as an example of Iberian crucible-based operations common during Chalcolithic times (see Chalcolithic metallurgy in the Northeast: between two technological traditions). Our microstructural analyses have confirmed the early appearance of smelting in the Northeast during the middle of the 3rd millennium BCE. Apart from using open vessels, the processing of a variety of regional oxidic ores in moderately reducing conditions would seem coherent with the southern Iberian technological tradition, although it is independent from it in terms of cultural connected spheres (see Chalcolithic metallurgy in the Northeast: between two technological traditions).

Smelting operations were developed across different occupational moments that gave different uses to space: from habitational contexts (levels III.1 and II.4) to a production context (level II.5). In the latter, production was higher, as deduced by the greater amount of crucible sherds recovered (see Materials analysed and their archaeological contexts). Because this level is between two very similar habitat settings, II.5 can be interpreted as a one-off moment in which this community needed to produce more copper. Production thus, was adapted to the generally low social demand that was usually fulfilled in domestic settings. These units integrated metallurgy in their other daily routines. Multiplying the number of metallurgical operations carried out in inefficient–but good enough–infrastructures (*i.e.* crucible metallurgy) when demand sporadically rises, rather than investing in improving efficiency to acquire more copper in a single operation, would remain a defining characteristic of Iberian metallurgy until IA times [83].

An adaptive strategy to demand fits better a utilitarian consideration of copper rather than a merely symbolic one, as a predominantly funerary use of metal would make more difficult to justify episodes of higher copper need such as the one reflected by level II.5. However, if it is

considered that utilitarian objects are still predominantly produced in other raw materials [29], then it can be seen how our sample characterisation complements conclusions extracted from analyses related to typology and depositional context. The panorama emerging is a metal valued by its utilitarian purposes (as reflected in the organisation of production that is adapted to this demand and in the exclusive manufacture of tool-weapons), but also of an incipient social value (perhaps emerging from its utilitarian superiority?), as demonstrated by its role as grave goods.

The smelting evidence and the apparent lack of recycling operations support availability and knowledge of copper resources to freshly produce metal. While our sample size is admittedly small, older metallographic analyses on other three further crucibles from La Bauma also identified them as smelting vessels. The absence of recycling may be related to the deposition of heavily used weapons in funerary contexts at this time (see The social value of Chalcolithic copper in the Western Mediterranean). It seems that when a metallic object reached the end of its usable life, it was deposited in tombs rather than recycled, and it acquired at that time a predominantly symbolic role. This behavioural pattern was sustainable because of the wide availability of superficial copper ore deposits in the Northeast [52].

The identification of the copper ores is informative of strategies for the exploitation of raw materials. The samples reflecting use of Ca-poor copper ores (E13 and F11) were uncovered in different strata that also contained samples associated to Ca-rich copper ores. These results, together with LIA corroborated that similar ores were processed in the context dedicated to metallurgical production and in the domestic contexts, indicating a versatile diachronic exploitation of different ore sources in both settings. Therefore, the change in the function of the site during the phase represented by level II.5 does not seem to imply a re-organisation of labour beyond multiplying metallurgical activities to meet higher demand.

The use of a minimum of five separate ore sources (see *supra*) within a single, small setting suggests a complex and versatile resource acquisition strategy in accordance to the highly mobile communities inhabiting La Bauma, and their lack of a centralised production organisation. Use of different copper ores since the early stages of metallurgy introduction shows a remarkable awareness of regional resources and a maintained network of connections with other groups that might have supplied some of these ores. This is especially possible for those mineral sources located at Montsant area (*ca*. 200km away) while other nearby sources such as those at Les Ferreres mining district (*ca*. 18km away) could have been directly exploited by the people that temporarily lived at La Bauma. This flexible and highly adaptative approach to resource acquisition would have ensured access to sufficient-enough amounts of raw materials, facilitating the development of the *ad hoc* metallurgy described so far for Bell Beaker communities.

Further characterisation and provenancing of the copper ores used in contemporaneous and later communities will help clarifying if these copper sources were communally exploited or if specific groups had priority access. Current evidence suggests a shared access to copper ore resources, perhaps consistent with the relatively low value of the metal. However, the difficulty for controlling access to the widely available copper resources should also be accounted for. The copper processed within the crucibles found at Balma del Duc and Cova del Buldó together with the copper used for manufacturing the awl from Cova de l'Heura seem to have originated from Turquesa mine (see Evidence of crucible metallurgy in the Northeast: Bauma del Serrat del Pont and its contemporaneous assemblages). Moreover, copper ore residues analysed from Cova de l'Heura were also linked to Montsant area (either Solana del Bepo or Barrac Fondo mines). It is still to be determined if the closer mining resources at Les Ferreres mining district were also shared between different groups. Preliminary evidence from finished

objects (see Lead isotopes) suggests so, although this should be confirmed with analyses of production residues, which are less likely to be subject to exchange.

The community at La Bauma used common pots for metallurgical activities, as typical of Iberian metallurgy. Even if some were decorated, this does not appear to have been an important consideration. This is again consistent with the mainly utilitarian metallurgy described for Bell Beaker groups.

It should be noticed that organic tempers/inclusions are not typically present in early Iberian crucibles in the South [11, 84–86] but they are common in other technological traditions [13]. For instance, organic temper was found in the non-movable crucible-*lingoteras* of La Capitelle du Broum [26]. Further analyses should confirm if the unintentional use of organic inclusions at La Bauma together with the use of organic temper in the closer French tradition, permeates into the technological tradition of the Northeast latter on as metallurgist realise the possibility of better preserving higher temperatures when crucibles are organic tempered vessels [87]. Interestingly, organic temper is used in crucibles from EBA sites in the upper Ebro basin, such as at El Abejar I (Navarra), Siete Cabezos and Moncín (both in Zaragoza), among others [88–90].

The technological study of samples from La Bauma exemplifies how microstructural, compositional and isotopic analyses of archaeometallurgical residues, when integrated in the socio-economic panorama of a particular case study, can be used as proxy to assess the social value of copper and copper metallurgy. Our data complements the previous work by Soriano Llopis [29] (see The social value of Chalcolithic copper in the Western Mediterranean) on typological studies, non-destructive compositional analyses, and stratigraphic contextualisation of materials related to copper metallurgy. Future technological studies of contemporaneous and later assemblages will help further assess the changing role of metallurgy in the Northeast.

The different trajectories for the social value of copper in Chalcolithic southeastern Iberia and France during the 2nd millennium BCE have been already discussed (see The social value of Chalcolithic copper in the Western Mediterranean). While the former is quickly involved in the development of elite power, in Languedoc, where metallurgy had played a more prominent social role in its origins, production and consumption stopped. It was advanced above that the process in the Northeast is yet different than for these two areas.

Full sedentarism, progressive nucleation of family units in bigger unwalled sites, and colonisation of the more fertile lower areas did not occur in the Northeast until the beginning of the 2nd millennium BCE at the inner plains of the Segre-Cinca Valley [46, 91–93]. This is already considered part of the next period in the area, the EBA-MBA (*ca*. 2300–1300 BCE). Sites such as Minferri (Juneda) (2050–1650 cal. BCE) and Cantorella (Maldà) (Fig 1) have considerable evidence of metallurgical production and are the best preserved examples of the so-called *agrupaciones de granjas dispersas* (groups of dispersed farmsteads) [47, 94–96]. Other areas of the Northeast maintained their cave-based mobile habitat patterns until the Late Bronze Age (LBA) [93].

It may be cautiously argued that during the beginning of the 2nd millennium BCE, an incipient social stratification emerged in the funerary record, but this is still not clear in material evidence from other contexts. This could have taken the form of 'big (wo)men': figures needed to coordinate labour and re-distribution of surpluses within these groups of farmsteads [29, 93, 94]. It is at these sites where Cu-based metallurgy (mainly tin bronze production now) appears fully developed in a considerable qualitative and quantitative step forward [47, 96–98]. After *ca*. 1600 BCE the social value of copper changed again, and ornaments are now found in the archaeological record associated to specific individuals. Different innovations related to metal production were also introduced (including new types of crucibles) as well as changes in

the organisation of production [29, 99]. Chiefdoms were not present in the Northeast until IA-I (750/650–550 BCE), and only in the Segre-Cinca Valley (e.g. Els Vilars) [91, 92, 100, 101]. State-like social structures were only developed after the emergence of the Iberian culture in the area during the Full Iberian period (*Ibérico Pleno*, 425/400–300 BCE).

Thus, three different trajectories can be seen in three very close areas. In the case of the Northeast and southern France, these were even culturally connected territories during the early phases of metallurgical production, but they later follow separate trajectories. This picture reinforces the idea of non-linearity towards social complexity once metallurgy is introduced and challenges unilinear explanations of technological development. These considerations are not only relevant for copper but research on the adoption of other technologies such as lead supports the importance of these kind of approaches [102]. Clearly, context-specific socio-political, ideological and environmental factors influencing the adoption and diffusion of innovations are indispensable to explain technological and social change [103, 104]. There is still work to do in order to better narrow down the key variables affecting change, a task only feasible if solid comparisons between different archaeological realities are established.

In comparison, in southeastern Iberia, the emerging elites that would give rise to El Argar culture used copper for boasting and consolidating their position (see The social value of Chalcolithic copper in the Western Mediterranean). In the Northeast, such elites did not develop so fast during the late $3^{rd}$ millennium BCE and the early $2^{nd}$ millennium BCE, owing to the social and economic dynamics of the area. Incipient social stratification stretches over time in the Northeast, and metallurgy (introduced in the middle of the $3^{rd}$ millennium BCE) enters a long process in which copper items compete with other technologies both in symbolic and utilitarian realms. This process extends until the MBA, when tin bronze replaces copper, and is followed by iron metallurgy during IA. Future studies should focus on the spread of these innovations and their integration within the on-going prestige system and subsistence economy. Comparative studies such as that by Kim [105] on the introduction of iron technology in Denmark and southern Korea demonstrate the potential of this kind of approaches. We moreover consider that integrating analytical studies in these approaches can enrich the data available for building well-grounded inferences.

## Supporting information

**S1 File. Metallography and SEM-EDS detailed results.**
(PDF)

**S2 File. Summary of the paper in Spanish / Resumen del artículo en castellano.**
(PDF)

**S3 File. Summary of the paper in Catalan / Resum de l'article en català.**
(PDF)

## Acknowledgments

We would like to thank Miquel Molist (Universitat Autònoma de Barcelona) and the staff of the Museu de la Garrotxa (Olot) for facilitating access to materials and sampling, and Catherine Kneale (McDonald Institute for Archaeological Research, University of Cambridge) and Simon Griggs (Department of Material Science and Metallurgy, University of Cambridge) for their technical support. The authors also thank SGIker (UPV/EHU/FEDER,EU) for technical and human support provided. We would also like to extend our gratitude to Marc Gener-Moret for his always useful advice, for facilitating the transport of some of the samples

involved in this research, and for the translation of the S3 File to Catalan. Jorge Canosa-Betés' guidance when creating Fig 1 is much appreciated. Finally, we thank the two anonymous reviewers for their feedback and comments.

## Author Contributions

**Conceptualization:** Julia Montes-Landa, Mercedes Murillo-Barroso.

**Formal analysis:** Julia Montes-Landa, Mercedes Murillo-Barroso, Ignacio Montero-Ruiz, Salvador Rovira-Llorens, Marcos Martinón-Torres.

**Funding acquisition:** Julia Montes-Landa, Mercedes Murillo-Barroso.

**Investigation:** Julia Montes-Landa, Mercedes Murillo-Barroso, Ignacio Montero-Ruiz, Salvador Rovira-Llorens.

**Methodology:** Julia Montes-Landa, Mercedes Murillo-Barroso, Ignacio Montero-Ruiz, Salvador Rovira-Llorens, Marcos Martinón-Torres.

**Project administration:** Julia Montes-Landa, Mercedes Murillo-Barroso.

**Resources:** Julia Montes-Landa, Mercedes Murillo-Barroso, Ignacio Montero-Ruiz, Salvador Rovira-Llorens, Marcos Martinón-Torres.

**Supervision:** Mercedes Murillo-Barroso, Marcos Martinón-Torres.

**Validation:** Julia Montes-Landa, Mercedes Murillo-Barroso, Ignacio Montero-Ruiz, Salvador Rovira-Llorens, Marcos Martinón-Torres.

**Visualization:** Julia Montes-Landa, Mercedes Murillo-Barroso.

**Writing – original draft:** Julia Montes-Landa, Mercedes Murillo-Barroso.

**Writing – review & editing:** Julia Montes-Landa, Mercedes Murillo-Barroso, Ignacio Montero-Ruiz, Salvador Rovira-Llorens, Marcos Martinón-Torres.

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
