## [Decision Letter · Decision Letter 0]

14 May 2021

PONE-D-21-13222

Interweaved traditions in Bell Beaker metallurgy: approaching the social value of copper at Bauma del Serrat del Pont (Northeast Iberia)

PLOS ONE

Dear Dr. Montes-Landa,

Thank you for submitting your manuscript to PLOS ONE. After careful consideration, we feel that it has merit but does not fully meet PLOS ONE’s publication criteria as it currently stands. Therefore, we invite you to submit a revised version of the manuscript that addresses the points raised during the review process.

All comments need to be addressed before re-submisison.

We look forward to receiving your revised manuscript.

Kind regards,

Peter F. Biehl, PhD

Academic Editor

PLOS ONE

Additional Editor Comments (if provided):

All comments (including language editing, e.g. title 'Interweaved') need to be addressed before re-submission.

Journal Requirements:

3) Please include captions for your Supporting Information files at the end of your manuscript, and update any in-text citations to match accordingly. Please see our Supporting Information guidelines for more information: http://journals.plos.org/plosone/s/supporting-information.

4)  We note that Figure 1 in your submission contain map/satellite images which may be copyrighted. All PLOS content is published under the Creative Commons Attribution License (CC BY 4.0), which means that the manuscript, images, and Supporting Information files will be freely available online, and any third party is permitted to access, download, copy, distribute, and use these materials in any way, even commercially, with proper attribution. For these reasons, we cannot publish previously copyrighted maps or satellite images created using proprietary data, such as Google software (Google Maps, Street View, and Earth). For more information, see our copyright guidelines: http://journals.plos.org/plosone/s/licenses-and-copyright.

i.    You may seek permission from the original copyright holder of Figure(s) [#] to publish the content specifically under the CC BY 4.0 license. 

ii.    If you are unable to obtain permission from the original copyright holder to publish these figures under the CC BY 4.0 license or if the copyright holder’s requirements are incompatible with the CC BY 4.0 license, please either i) remove the figure or ii) supply a replacement figure that complies with the CC BY 4.0 license. Please check copyright information on all replacement figures and update the figure caption with source information. If applicable, please specify in the figure caption text when a figure is similar but not identical to the original image and is therefore for illustrative purposes only.

Reviewers' comments:

Reviewer's Responses to Questions

**Comments to the Author**

1. Is the manuscript technically sound, and do the data support the conclusions?

Reviewer #1: Yes

Reviewer #2: Yes

2. Has the statistical analysis been performed appropriately and rigorously? 

Reviewer #1: N/A

Reviewer #2: N/A

3. Have the authors made all data underlying the findings in their manuscript fully available?

Reviewer #1: Yes

Reviewer #2: Yes

4. Is the manuscript presented in an intelligible fashion and written in standard English?

Reviewer #1: Yes

Reviewer #2: Yes

5. Review Comments to the Author

Reviewer #1: - The paper, after a useful review of the ideas concerning the differences in the technologies and social value of copper, presents original research based on a study of 7 crucible shards out of 65 from La Bauma. Despite the small number of samples the maximum information is extracted using ED-SEM and LIA analyses with comparisons with other archaeological sites and possible ore sources.

- The paper is generally well written and presents results not published elsewhere. The analyses are presented to a high technical standard.

- All the comments listed below are minor and mainly suggest some points that need a little clarification:

- p.6. Is the suggestion that all/most metalwork from this region is early tin-bronze (possibly by natural ore Cu-Sn mixtures) or is it just a few exceptional items?

- Line 96. As the paper is based on 7 crucible fragment a more detailed description or even a drawing of the shape of crucible would be useful as many readers will picture a modern cup crucible while yours sounds more like the flat dish type.

- p.23. The very interesting observation regarding the elevated CaO levels suggests, as you note, the presence of calcite (limestone) or dolomite as a gangue mineral. You could check the geology of the ores at your possible sites. Malachite (copper carbonate) is most stable and abundant when copper deposits occur in limestones or dolomites (alkaline) and is rather scarce in more acid conditions. Hence, the question is , do any of your ore sources occur in limestones or dolomites? This might strengthen your provenance case ( see Williams, R.A. (2014) Linking Bronze Age copper smelting slags from Pentrwyn on the Great Orme to ore and metal. In: The origins of metallurgy in Europe. Historical Metallurgy 47(1) pp.93-110).

- p.25. You mention delafossite indicating poorly reducing conditions. You make no mention of the absence of more typical phases seen in copper smelting such as fayolite, wustite etc. However, I presume you are suggesting as the copper was obtained from smelting from very pure ore (and prills recovered by crushing any small amounts of slag) there was no need for iron and silica levels to form copper smelting slags (see reference cited above).

- p.26. You mention low levels of As, Ag and Sn. However, the levels you quote are near the detection levels for ED-SEM ([particularly arsenic which has a higher detection level than most elements). If this is true, mentioning the detection limits (or that you are near to them) would be advisable.

- p28. Perhaps make it clear earlier exactly what you have analysed for lead isotopes (i.e. Five slag samples). In Figure 19 a better description of the ‘geological’ samples are ‘copper ores’ and the ‘archaeological’ samples as ‘slags’.

- p.36. line 860 and line 885. Please clarify comments about tin-bronze use and dates in each area as its not entirely clear.

Overall, a very good paper and a useful addition to the field of research.

Reviewer #2: The manuscript provides a very welcome and thorough contextualisation of the data from early NE Spanish crucibles vis-a-vis the neighbouring regions in France and SE Spain. The literature review is thorough and brings out the relevant systematic difference sin metallurgical practice, laying the foundation for the subsequent new data to be interpreted. The data presentation and interpretation are done well, with substantive and necessary micrographs, clear SEM-EDS analyses, and descriptions.

This reviewer would, however, recommend that the authors look a little more critically into the LIA data and their interpretation, given that they analyse crucible slag which is mostly based on fused ceramic from the vessel, fuel ash from the internal heating, and whatever charge component was trapped in the sticky slag. Firstly, it would be necessary to consider the lead concentrations in the various primordial materials: ceramic, fuel ash, and gangue. My impression is that the ore / metal here is generally low in lead, which makes it imperative to consider the three potential contributing source materials to the resulting LIA signature of the slag. Rademakers et al (2017) in JAS have done this recently for LBA bronze melting crucibles from Egypt, and have been able to tease out the distorting effect of the ceramic contribution to the overall LIA signature. A similar critical evaluation of the data here is strongly recommended. If no trace element lead data is available, then at least the authors need to acknowledge this, and discuss the possibility of contamination / mixing, rather than going straight into the provenance discussion and wonder about mis-match of slag LIA and potential ore source LIA signatures. (And they should explain why [line 737] the unknown ore source 'is probably within the region'.)

Both in the Introduction and the Discussion the authors refer to three previously-analysed crucibles from La Bauma, but without giving references: are these from within the author team? If not, might it be possible to give more information regarding this earlier data?

Overall, the emerging picture from this study is fascinating in the diversity of ore source exploitation at the beginning of extractive metallurgy, and the long-range networked nature of the communities involved. Exploiting ore sources several hundred km away in an ad hoc utilitarian household metallurgy, when nearer sources are evidently being used as well, is intriguing. This, and the potential exploitation of a complex Cu-Sn ore as the fifth source [lines 740-745] to smelt a natural tin bronze makes this scenario remarkably similar to the picture emerging from the 5th millennium Balkans, as recently detailed by Radivojevic and co-workers in several publications, even though there seem to be no crucibles used in that region. Notwithstanding the three distinct trajectories for metallurgical practice identified in this study, one might wonder whether the multi-source, long-range networked and dispersed organisation (if this is the right word) of metallurgical practice is an inherent feature of emerging metallurgies, where a multitude of village / mobile smelters operate across a wide landscape and exchange experience as much as material and practices while maintaining diversity, before at some point in the Bronze Age a more focussed and standardised large-scale production model of copper smelting and aactive alloying with tin metal replaces the almost anarchic earlier situation.

The first word in the title should be 'Interwoven' and not 'Interweaved'

6. PLOS authors have the option to publish the peer review history of their article (what does this mean?). If published, this will include your full peer review and any attached files.

Reviewer #1: No

Reviewer #2: No

---

## [Author Response · Author response to Decision Letter 0]

6 Jul 2021

Journal Requirements:

Please review your reference list to ensure that it is complete and correct. Any changes to the reference list should be mentioned in the rebuttal letter that accompanies your revised manuscript. 

New references incorporated: 

1. When asked to clarify the presence/absence of fayalite/wüstite by Reviewer 1 (see below), the following reference was incorporated: 

a. Müller R, Goldenberg G, Bartelheim M, Kunst M, Pernicka E. Zambujal and the beginnings of the metallurgy in southern Portugal. In: La Niece S, Hook D, Craddock P, editors. Metals and Mines Studies in Archaeometallurgy Selected Papers from the Conference “Metallurgy: a Touchtone for Cross-cultural Interaction” Held at the British Museum 28-30 April 2005 to Celebrate the Career of Paul Craddock During his 40 Years at the British Museum. London: Archetype Publications; 2007. pp. 15–26. 

2. We incorporated the following references to further support that early Iberian crucibles from the South do not usually contain organic temper: 

a. Mongiatti A, Montero-Ruiz I. Rediscovering famous assemblages: a rare Bronze Age crucible from El Argar, Spain. Archaeometry. 2020; 62: 329–345.

b. del Pino Curbelo M, Day PM, Camalich Massieu MD, Martín Socas D, Molina González F. Plus ça change: pots, crucibles and the development of metallurgy in Chalcolithic Las Pilas (Mojácar, Spain). Archaeological and Anthropological Sciences. 2019;11: 1553–1570.

c. Hook DR, Freestone IC, Meeks ND, Craddock PT, Moreno Onorato A. The early production of copper alloys in Southeast Spain. In: Pernicka E, Wagner GA, editors. Archaeometry ’90. Birkhauser Verlag; 1991. pp. 65–76.

3. The following references were incorporated in Fig 19 caption to clarify the sources of the data used in these graphs:

a. Montero-Ruiz I, Gener M, Renzi M, Hunt M, Rovira S, Santos-Zalduegui JF. Provenance of the lead in First Iron Age sites in Southern Catalonia (Spain). In: Moreau JF, Auger R, Chabot J, Herzog A, editors. Proceedings ISA 2006, 36th International Symposium on Archaeometry, 2-6 May 2006, Quebec City, Canada. Quebec: Laboratoire d’archéologie, Université du Québec à Chicoutimi; 2009.

b. Rafel N, Hunt Ortiz M, Montero Ruiz I, Soriano I, Delgado-Raack S, Marín D. New Bronze Age absolute datings for Solana del Bepo copper mine (Ulldemolins, Tarragona province, Spain). Mediterranean Archaeology and Archaeometry. 2019;19: 9–24.

4. According to Reviewer 2 comment (see below), we added the following reference when discussing presence of Pb in the ceramic paste of the crucibles: 

a. Rademakers FW, Rehren T, Pernicka E. Copper for the Pharaoh: Identifying multiple metal sources for Ramesses’ workshops from bronze and crucible remains. Journal of Archaeological Science. 2017;80: 50–73.

5. When discussing the comment of Reviewer 1 in relation to Ca levels supporting LIA characterisation, the following references were added: 

a. Andreazini A, Melgarejo JC, Rafel Fontanals N, Soriano I. The structure and mineralogy of the mine. In: Rafel Fontanals N, Hunt Ortiz MA, Soriano I, Delgado-Raack S, editors. Prehistoric copper mining in the northeast of the Iberian Peninsula: La Turquesa o Mas de les Moreres Mine (Cornudella de Montsant, Tarragona, Spain). Lleida: Universidad de Lleida; 2018. pp. 24–32.

b. Delgado-Raack S, Gómez-Grass D. Technological-functional study of the macrolithic artefacts from Solana del Bepo. Revista d’Arqueologia de Ponent. 2017;2: 45–63.

Minor changes were done on the first page including appropriate indication of the corresponding author and removal of the short title. We have not used symbols to show the different contributions from each author because there are more than two levels of contributions. We hope the author’s contributions information provided according to CRediT standards can help to clarify how work was distributed. We would like our contributions to be recorded following this way if possible. We have maintained the section with the different ORCiD numbers because we would like to have these included in the paper published. There was no space to provide the ORCiD numbers of the authors (except for the corresponding author) during submission. 

The format of each section and sub-section headings was amended and their numbering removed. In-text references to other sections/sub-sections were amended accordingly.

The title of the figure captions was made bold, justified and italics were undone. The title of the tables was made bold, justified and italics were undone. The legend was moved to the lower part when relevant. 

Please include captions for your Supporting Information files at the end of your manuscript, and update any in-text citations to match accordingly. 

A ‘Supporting information’ section was added at the end of the document listing the supplementary materials provided. “Supplementary materials” was substituted on the text with “supporting information” and direct reference to the S1 File was incorporated (second line of the results section).

S2 and S3 files were added. They are short summaries of the paper in Spanish and Catalan. The purpose of these additions is to make this paper accessible to the people of the area of research in their native languages. Captions were provided for these files too.

We note that Figure 1 in your submission contain map/satellite images which may be copyrighted. All PLOS content is published under the Creative Commons Attribution License (CC BY 4.0), which means that the manuscript, images, and Supporting Information files will be freely available online, and any third party is permitted to access, download, copy, distribute, and use these materials in any way, even commercially, with proper attribution. For these reasons, we cannot publish previously copyrighted maps or satellite images created using proprietary data, such as Google software (Google Maps, Street View, and Earth). 

The data of the raster model used on Fig 1 comes from a public domain source (SRTM digital elevation model from NASA), so no copyright permit should be needed. Credits were added on Fig 1 caption to acknowledge this.

Reviewer 1 comments: 

p.6. Is the suggestion that all/most metalwork from this region is early tin-bronze (possibly by natural ore Cu-Sn mixtures) or is it just a few exceptional items?

The phrasing of the sentence indicated by the reviewer was somewhat confusing because in the previous sentences, we discuss together both Chalcolithic and EBA evidence at La Bauma. Thus, we moved this sentence to start a different paragraph with it. We also added the world “Chalcolithic” at the beginning of the paragraph to further emphasise that we are referring to Chalcolithic evidence.

Chalcolithic samples from La Bauma are the earliest evidence of extractive metallurgy in the area (discussed in the previous paragraphs). Previous compositional analyses of the Chalcolithic samples suggest that early smelting activities involved both copper smelting and bronze production through a natural alloying process. This was clarified in this paragraph.

We would like to emphasise that here we are discussing the beginning of extractive metallurgy (as mentioned in the sentence) and not the first metalwork items, as the reviewer seems to understand. Metallic copper (not tin bronze) items were present in the Northeast before extractive metallurgy was conducted, introduced from France. This is explained and discussed on the section dedicated to ‘The social value of Chalcolithic copper in the Western Mediterranean’. We changed the world “used” by “produced” to further clarify this.

The resulting paragraph with all changes (underlined) applied follows:

“The analyses of Chalcolithic samples from La Bauma, highlight that the beginning of extractive metallurgy in the Northeast involves both copper smelting and bronze production through a potential natural alloying process. This contrasts with southern Iberia, where pure copper and/or arsenical copper was produced. In France, some analyses of early daggers also point towards an early introduction of bronze metallurgy [38].”

Line 96. As the paper is based on 7 crucible fragment a more detailed description or even a drawing of the shape of crucible would be useful as many readers will picture a modern cup crucible while yours sounds more like the flat dish type.

Line 96 in the original version is in italics: “Analyses of assemblages from these sites show that Iberian Chalcolithic communities conducted a simple crucible-based metallurgy developed within vasijas-horno (‘crucible-furnaces’): shallow open common pottery vessels with rounded bottoms; very different from other Neolithic/Chalcolithic crucibles from Europe and the Near East and from modern cup crucibles [1].”

The reference at the end of the sentence provides comparisons of Iberian crucibles and different European and Near East ones, including pictures. We added the underlined clarification (see above) at the end of the sentence to avoid the confusion with cup crucibles.

It is also clarified latter on that the crucibles used were common pots, some of them with decorations, what should debunk the conception of picturing a modern cup crucible:

“The community at La Bauma used common pots for metallurgical activities, as typical of Iberian metallurgy. Even if some were decorated, this does not appear to have been an important consideration.”

Fig 2 provides pictures of the sherds analysed. There are no rims or bottoms that allow to accurately reconstruct the shape of these vessels. Some of the crucibles found at La Bauma with preserved rims (not the ones here analysed) were drawn by Alcalde et al. (1998). We have incorporated the following sentence (underlined) to direct the interested readers to this publication already mentioned in the original text: 

“The 65 crucible sherds found belong to a minimum of 14 undecorated vessels and five decorated Bell Beaker pots. On Alcalde et al. [2] drawings of some of these vessels are available. The use of decorated pots for metallurgical operations is not a unique feature of La Bauma.”

p.23. The very interesting observation regarding the elevated CaO levels suggests, as you note, the presence of calcite (limestone) or dolomite as a gangue mineral. You could check the geology of the ores at your possible sites. Malachite (copper carbonate) is most stable and abundant when copper deposits occur in limestones or dolomites (alkaline) and is rather scarce in more acid conditions. Hence, the question is , do any of your ore sources occur in limestones or dolomites? This might strengthen your provenance case ( see Williams, R.A. (2014) Linking Bronze Age copper smelting slags from Pentrwyn on the Great Orme to ore and metal. In: The origins of metallurgy in Europe. Historical Metallurgy 47(1) pp.93-110).

Calcium rich minerals are common in the Iberian NE but, as the reviewer notes, this element can strengthen the provenance assignation.

Calcite is common at the area of the Pyrenees around La Bauma. However, the mineralisation exploited at Les Ferreres mine –the possible local source– is surrounded by granite. This does not contradict the isotopic characterisation of the sample related to Les Ferreres mine, as this sample had no calcium enrichment. 

Samples related to La Turquesa mine and Solana del Bepo were richer in calcium and calcium minerals are common at both mines: Crandallite (CaAl3(PO4)(PO3OH)(OH)6) is the most common mineral in the superficial zone of the deposit at La Turquesa mine, and there are also small quantities of fluorapatite (Ca5PO4)3F) (Andreazini et al. 2018). Solana del Bepo is a Ba-Cu vein with calcite associated. Its geological context is largely made up of slates affected to a greater or lesser extent by contact metamorphism from the Carboniferous-Permian, above which, locally, are limestones of the Cornudella Group from the Palaeocene-Eocene (Delgado-Raack and Gómez-Gras 2017: fig. 27) therefore it would not be surprising that some calcium rich mineral entered the charge with the copper ores. This information has been added in the section ‘Lead Isotopes’.

p.25. You mention delafossite indicating poorly reducing conditions. You make no mention of the absence of more typical phases seen in copper smelting such as fayolite, wustite etc.

No fayalite and wüstite were found in this assemblage. The copper ores used had low amounts of impurities. As indicated in the ‘Results’ sections, in some samples, semi-dissolved Fe-rich minerals related to the decomposition of the ceramic paste were found, so crucible decomposition contributed some iron to the system. Crucible smelting in open vessels imposes a variable reducing atmosphere (evidenced by the formation of delafossite and cuprite). This makes fayalite development complicated (although this might happen in some cases as the reviewer points out). Early copper smelting evidence from Iberia do not commonly present fayalite and wüstite, so this should not be surprising. 

The following sentence (underlined) was added in the relevant paragraph dealing with iron presence: 

“In addition to CaO and MgO, the high levels of FeO in some neo-silicates should be discussed. In sample F12, individual crystals contain up to 37.5wt%FeO. In H12, some bright clusters of Fe silicates can be identified (Fig 11). However, given that no copper is associated to these clusters and that similar mineral inclusions are recognised in the ceramic paste of this vessel (Table 4), it is plausible that these slag phases and the overall FeO enrichment derive, at least in part, from semi-dissolved ceramic inclusions. Thus, it is difficult to confidently discern the extent to which Fe was part of the crucible charge. Fayalite and wüstite phases were not found in any sample, in agreement with other early smelting evidence from Iberia [11,75].”

The two references provided exemplify the common absence of these phases in Iberian early metallurgy. The former was already mentioned somewhere else on the text, and the latter was added to attend the reviewer’s comment and incorporated to the reference list (see edited reference list above).

However, I presume you are suggesting as the copper was obtained from smelting from very pure ore (and prills recovered by crushing any small amounts of slag) there was no need for iron and silica levels to form copper smelting slags (see reference cited above).

The process mentioned by the reviewer is typical of early Iberian metallurgy and is explained on the following paragraph of the section dedicated to ‘Chalcolithic metallurgy in the Northeast: between two technological traditions’. This paragraph already includes references to Iberian cases. The use of very pure ores is mentioned, implying that no conscious Fe and/or Si addition was intended to produce smelting slag:

“Analyses of assemblages from these sites show that Iberian Chalcolithic communities conducted a simple crucible-based metallurgy developed within vasijas-horno (‘crucible-furnaces’): shallow open common pottery vessels with rounded bottoms; very different from other Neolithic/Chalcolithic crucibles from Europe and the Near East and from modern cup crucibles [13]. Local copper carbonates and oxides were smelted under variable reducing conditions that usually produced an incomplete reaction of the charge [14–17]. These processes generated no or little slag and when they did, this was crushed to recover the metallic prills trapped. As is common in early metallurgy, a metallic mass of the desired shape was cast after re-melting the prills, and was later hammered [17,18]. Contrary to other parts of Europe, annealing was not extensively used in Iberia until later times, so objects were usually cold hammered [10].”

Ore impurities are described in detail on the ‘Results’ section. Here, the sources of Fe (from ceramic paste and possibly from some minor ore impurities) and Si (mainly the decomposition of the ceramic paste) are discussed, so no relation to fluxes is identified for these elements. The composition of the ores used is also summarised at the beginning of the ‘Discussion’ section.

The following sentence located latter in the Discussion section, references back to the first paragraph here quoted, thus relating these samples to the Iberian Chalcolithic tradition described: 

“These results allow to technologically classify La Bauma metallurgy as an example of Iberian crucible-based operations common during Chalcolithic times (see Chalcolithic metallurgy in the Northeast: between two technological traditions)”.

We consider that all this already makes clear the type of ores used, their impurities, the unmatured nature of the slag, and the materials contributing to its formation (especially the relationship of Fe and Si with the decomposition of the ceramic paste and not the addition of fluxes). 

p.26. You mention low levels of As, Ag and Sn. However, the levels you quote are near the detection levels for ED-SEM (particularly arsenic which has a higher detection level than most elements). If this is true, mentioning the detection limits (or that you are near to them) would be advisable.

The following underlined information was added: 

“Moreover, some prills in G11 present minor As (<0.6wt%As) and Ag (<0.6wt%Ag). A single prill was found to contain 0.4wt%As and 0.6wt%Sn. Interestingly, no prill combines the three impurities (As, Ag and Sn), although all of them have some Fe. However, it is necessary to note that the results reported for these minor elements are close to the detection limits of the SEM-EDS (~0.1-0.3wt%).”

p28. Perhaps make it clear earlier exactly what you have analysed for lead isotopes (i.e. Five slag samples).

This was made clearer on the ‘Methods‘ section when introducing LIA, and at the very beginning of the ‘Lead isotopes’ section, where these results are discussed.

In Figure 19 a better description of the ‘geological’ samples are ‘copper ores’ and the ‘archaeological’ samples as ‘slags’.

Not all geological samples are copper ores. Cu or Pb has been added to the legend, and references to the original publication of this data have been included. These papers offer broader geological information of the mineralisations and samples analysed. 

Similarly, not all archaeological samples are slags. ‘S’ or ‘M’ has been added to the legend to indicate the analysis of ‘Slag’ or ‘Metals’. References to the original publications of this data were added. Descriptions of the archaeological samples used in these graphs are also included in the section ‘Evidence of crucible metallurgy in the Northeast: Bauma del Serrat del Pont and its contemporaneous assemblages’.

p.36. line 860 and line 885. Please clarify comments about tin-bronze use and dates in each area as its not entirely clear.

The references provided at the end of the sentence summarise the EBA-MBA evidence related to tin bronze production in the region. Given that this paper is focused on copper metallurgy and not on tin bronze one (a technology that we do not discuss in detail at any moment in this paper), we do not consider that further discussion on tin bronze is necessary here and that we have already provided the references to guide interested readers. The paragraph where this line is, gives a general overview of what was happening in the Northeast from EBA-MBA until IA, in order to briefly exemplify how the trajectory towards social complexity differs from Southern Iberia and Southern France ones. We believe that this is already fulfilled with the current data and references discussed.

Dates are already provided on the previous paragraph (“This is already considered part of the next period in the area, the EBA-MBA (ca. 2300-1300 BCE)”), and at the beginning of the paragraph where the original line 860 is located (“…during the beginning of the 2nd millennium BCE...”).

The paragraph where the referenced lines are located and the previous one, are focused on the Northeast. No different areas are considered, as the reviewer pointed out. On the previous paragraph, we clarified that the Segre-Cinca Valley is located within the Northeast.

Reviewer 2 comments: 

This reviewer would, however, recommend that the authors look a little more critically into the LIA data and their interpretation, given that they analyse crucible slag which is mostly based on fused ceramic from the vessel, fuel ash from the internal heating, and whatever charge component was trapped in the sticky slag. Firstly, it would be necessary to consider the lead concentrations in the various primordial materials: ceramic, fuel ash, and gangue. My impression is that the ore / metal here is generally low in lead, which makes it imperative to consider the three potential contributing source materials to the resulting LIA signature of the slag. Rademakers et al (2017) in JAS have done this recently for LBA bronze melting crucibles from Egypt, and have been able to tease out the distorting effect of the ceramic contribution to the overall LIA signature. A similar critical evaluation of the data here is strongly recommended. If no trace element lead data is available, then at least the authors need to acknowledge this, and discuss the possibility of contamination / mixing, rather than going straight into the provenance discussion and wonder about mis-match of slag LIA and potential ore source LIA signatures. 

We agree with the reviewer on the fact that this question worth to be analysed further. For this paper we lack trace element composition of the ceramic paste or of the potential clay sources around the site and therefore an evaluation of the potential contamination by the ceramic paste is not possible. However, the compositional analyses of copper ores from the mining districts considered show high levels of lead, some of them being Pb-Cu ores (up to 70% Pb). Therefore, assuming that this kind or ore is smelted here, the main Pb contribution in the slag formation would be that of the copper ore and not the ceramic paste. However, for future investigations we will consider analysing Iberian ceramic pastes to evaluate if lead rich clays are used in the different regions. This comment has been addressed in the ‘Lead isotope’ section with the following paragraph, that also includes the reference mentioned by the reviewer: 

“Regarding provenance and considering that we are dealing with slag layers here, one should consider the potential lead contribution from the three main sources of slag formation: i.e. the ceramic paste, the copper ore and the fuel. Rademakers et al. [76] have shown that crucibles made of Pb-rich silt (20-1500 ppm Pb in the ceramic paste) can affect lead isotopic compositions of their slags layers. Nonetheless, this contamination is expected to be minor when Pb-poor clay is used and especially if lead-rich ores are smelted. For the case of La Bauma, most of the copper ores known in the Northeast and considered as potential sources here (except for La Turquesa mine) have significant levels of lead, some of them being actually Cu-Pb minerals [54,58]. Therefore, despite lacking trace element composition of the ceramic pastes or surrounding clays, it is assumed that the main lead contribution to the slag will be that of the copper ore, and therefore it would be possible to match La Bauma samples with the regional mining districts.”

And they should explain why [line 737] the unknown ore source 'is probably within the region'.

We had provided an explanation for this on the ‘Lead isotopes’ section: 

“…Although no specific provenance can be proposed with the available data, these samples follow the same trend observed for all the northeastern metallurgical debris analysed so far. It seems possible that some regional resources (but still different ones than the characterised up to now) could have been used in these cases as well, as there are still some lacunas of isotopic information related to some mining districts in the area.”

We have clarified this at the end of the sentence pointed out by the reviewer (added part underlined): 

“However, this unknown copper source is probably within the region, as the analysed samples from La Bauma follow the same trend than previously analysed samples from the Northeast (Fig 18, see Lead isotopes).”

This addition refers to the previous quoted paragraph and to the following sentence on the ‘Lead isotopes’ section, where also the underlined clarification was added:

“Interestingly, samples from La Bauma follow the same trend than previous northeastern samples analysed, filling an existing gap (Fig 18), perhaps indicating that all minerals used originated in this region.”

Both in the Introduction and the Discussion the authors refer to three previously-analysed crucibles from La Bauma, but without giving references: are these from within the author team? If not, might it be possible to give more information regarding this earlier data?

On the section dedicated to ‘Chalcolithic metallurgy in the Northeast: between two technological traditions’, it was specified that these crucibles were analysed by S. Rovira, one of the authors:

“…which were complemented by some metallographic analyses by S. Rovira (unpublished).” 

We also provide information about their contextual data on the ‘Materials’ section, but the underlined text was added to facilitate cross-references:

“Before the study of these seven crucibles was conducted, metallographies of another three crucibles from La Bauma had been conducted but never published (see Chalcolithic metallurgy in the Northeast: between two technological traditions). With the aim of making these older results available, a short description of these older micrographs is provided below (see Results) in relation to the newer analyses. The references for these samples are PA6325, PA6326 and PA6327.”

(…)

“The information available for the older, previously unpublished, samples studied by S. Rovira positions them in layers II.4 and II.5, supporting a chronological relationship with the rest of the assemblage analysed.”

Our paper summarises this original data previously unavailable at the end of the ‘Metallurgical operations with Ca-rich charges’ sub-section (two last paragraphs), including the metallographies available (Figures 13, 14 and 15). We think that all the information is already on the text.

Overall, the emerging picture from this study is fascinating in the diversity of ore source exploitation at the beginning of extractive metallurgy, and the long-range networked nature of the communities involved. Exploiting ore sources several hundred km away in an ad hoc utilitarian household metallurgy, when nearer sources are evidently being used as well, is intriguing. This, and the potential exploitation of a complex Cu-Sn ore as the fifth source [lines 740-745] to smelt a natural tin bronze makes this scenario remarkably similar to the picture emerging from the 5th millennium Balkans, as recently detailed by Radivojevic and co-workers in several publications, even though there seem to be no crucibles used in that region. Notwithstanding the three distinct trajectories for metallurgical practice identified in this study, one might wonder whether the multi-source, long-range networked and dispersed organisation (if this is the right word) of metallurgical practice is an inherent feature of emerging metallurgies, where a multitude of village / mobile smelters operate across a wide landscape and exchange experience as much as material and practices while maintaining diversity, before at some point in the Bronze Age a more focussed and standardised large-scale production model of copper smelting and active alloying with tin metal replaces the almost anarchic earlier situation.

We agree on this interesting point; we share the reviewer’s view. Parallels with Balkan metallurgy are suggestive but given that the intentionality of tin bronze production at La Bauma during Chalcolithic times still needs to be further assessed, we would not like to include further discussion on bronze metallurgy in this paper, focused on copper metallurgy, as it would open a completely different set of questions that are beyond the scope of this paper (from intentionality of early bronze production in the Northeast, to exploration of the origin of this innovation, and consequences in social complexity development latter on, etc.). We consider that our results focused on Chalcolithic copper metallurgy solidly stand by their own and that enough information on tin bronze metallurgy and sufficient references to relevant sources are provided to guide interested readers. The point raised by the reviewer deserves further attention in the future.

The first word in the title should be 'Interwoven' and not 'Interweaved'.

Corrected.

Other minor editing changes: 

• Space at both sides of “=” symbols were added for consistency. 

• The two anonymous reviewers were added to the acknowledgements.

• All the “et al.” mentioned on the text were italised.

• n-dashes between dates were changed for m-dashes.

• “Mid-3rd millennium BCE” and similar constructions were replaced by “middle of the 3rd millennium BCE”.

• “Anal. Total” was substitute by “Analytical Total”.

• We have substituted BC acronymic with BCE.

• Some verb tenses were changed to past tenses for consistency.

• Other minor correction of typos or slight re-phrasing of sentences can be seen in the tracked changes version.

• We produced a new version of Fig 1. Now the map separates more clearly archaeological sites from mines and it incorporates a legend. The caption for Fig 1 was edited accordingly.

• Fig 18 and Fig 19 were edited according to the reviewer comments (see above).

• We have done minor edits (all related to style and not content) in the S1 File. This was now uploaded as a PDF.

---

## [Decision Letter · Decision Letter 1]

26 Jul 2021

Interwoven traditions in Bell Beaker metallurgy: Approaching the social value of copper at Bauma del Serrat del Pont (Northeast Iberia)

PONE-D-21-13222R1

Dear Dr. Montes-Landa,

We’re pleased to inform you that your manuscript has been judged scientifically suitable for publication and will be formally accepted for publication once it meets all outstanding technical requirements.

Kind regards,

Peter F. Biehl, PhD

Academic Editor

PLOS ONE

Additional Editor Comments (optional):

Reviewers' comments:

Reviewer's Responses to Questions

**Comments to the Author**

1. If the authors have adequately addressed your comments raised in a previous round of review and you feel that this manuscript is now acceptable for publication, you may indicate that here to bypass the “Comments to the Author” section, enter your conflict of interest statement in the “Confidential to Editor” section, and submit your "Accept" recommendation.

Reviewer #1: All comments have been addressed

Reviewer #2: All comments have been addressed

2. Is the manuscript technically sound, and do the data support the conclusions?

Reviewer #1: (No Response)

Reviewer #2: Yes

3. Has the statistical analysis been performed appropriately and rigorously? 

Reviewer #1: (No Response)

Reviewer #2: N/A

4. Have the authors made all data underlying the findings in their manuscript fully available?

Reviewer #1: (No Response)

Reviewer #2: Yes

5. Is the manuscript presented in an intelligible fashion and written in standard English?

Reviewer #1: (No Response)

Reviewer #2: Yes

6. Review Comments to the Author

Reviewer #1: (No Response)

Reviewer #2: Thank you for implementing some of the suggestions, and for justifying why not the others - all good!

7. PLOS authors have the option to publish the peer review history of their article (what does this mean?). If published, this will include your full peer review and any attached files.

Reviewer #1: No

Reviewer #2: No

---

## [Editor Report · Acceptance letter]

30 Jul 2021

PONE-D-21-13222R1 

Interwoven traditions in Bell Beaker metallurgy: Approaching the social value of copper at Bauma del Serrat del Pont (Northeast Iberia) 

Dear Dr. Montes-Landa:

I'm pleased to inform you that your manuscript has been deemed suitable for publication in PLOS ONE. Congratulations! Your manuscript is now with our production department. 

Kind regards, 

on behalf of

Dr. Peter F. Biehl 

Academic Editor

PLOS ONE